# Dual roles and evolutionary implications of P26/poxin in antagonizing intracellular cGAS-STING and extracellular melanization immunity

Mengyi Yin [1,2,5], Wenhua Kuang[1,5], Qianran Wang[1], Xi Wang[1], Chuanfei Yuan[1], Zhe Lin[3], Huanyu Zhang[1], Fei Deng [1], Haobo Jiang [4], Peng Gong [1] ✉, Zhen Zou [2,3] ✉, Zhihong Hu [1] ✉ & Manli Wang [1] ✉

P26, a homolog of the viral-encoded nuclease poxin that neutralizes the cGAS-STING innate immunity, is widely distributed in various invertebrate viruses, lepidopteran insects, and parasitoid wasps. P26/poxin from certain insect viruses also retains protease activity, though its biological role remains unknown. Given that many P26s contain a signal peptide, it is surmised that P26 may possess certain extracellular functions. Here, we report that a secretory baculoviral P26 suppresses melanization, a prominent insect innate immunity against pathogen invasion. P26 targets the cofactor of a prophenoloxidase-activating protease, and its inhibitory function is independent of nuclease activity. The analysis of P26/poxin homologs from different origins suggests that the ability to inhibit the extracellular melanization pathway is limited to P26s with a signal peptide and not shared by the homologs without it. These findings highlight the independent evolution of a single viral suppressor to perform dual roles in modulating immunity during virus-host adaptation.

The cytosolic cGAS-STING pathway is part of an important innate immune response for antagonizing DNA virus infection. Recently, an immune nuclease poxin was identified in poxviruses to inhibit this cytosolic DNA-sensing pathway by selectively degrading 2'3'-cyclic GMP-AMP (cGAMP)[1]. *Poxin* homologs widely exist in the genome of invertebrate DNA and RNA viruses, as well as in hymenopteran parasitoid wasps and lepidopteran insects[1], suggesting horizontal gene transfer events between insect hosts and their viruses/parasitoids, and further radiation to mammalian viruses. Structural and functional analyses suggest that poxins may have evolved from viral proteases through a gain of secondary nuclease activity[2]; however, the biological role of these viral proteases remains unknown. In addition, many poxin homologs contain a signal peptide, implying an extracellular function in addition to counteracting the cytosolic cGAS-STING pathway.

The homolog of poxin was initially identified in insect viruses of *Baculoviridae* and called P26, because it encodes a viral protein with a molecular mass of 26 kDa[3]. The detailed role of P26 during baculovirus infection remains unclear[4-6]. In the parasitoid wasps *Microplitis mediator* and *Microplitis demolitor*, P26 is secreted and highly abundant in venom, which is introduced into lepidopteran hosts during wasp oviposition to suppress host immune response[7,8]. Prediction of the signal peptide and phylogenetic analysis suggest that secreted P26/poxin

---

[1]State Key Laboratory of Virology, Wuhan Institute of Virology, Chinese Academy of Sciences, Wuhan, China. [2]University of Chinese Academy of Sciences, Beijing, China. [3]State Key Laboratory of Integrated Management of Pest Insects and Rodents, Institute of Zoology, Chinese Academy of Sciences, Beijing, China. [4]Department of Entomology and Plant Pathology, Oklahoma State University, Stillwater, OK, USA. [5]These authors contributed equally: Mengyi Yin, Wenhua Kuang. ✉e-mail: gongpeng@wh.iov.cn; zouzhen@ioz.ac.cn; huzh@wh.iov.cn; wangml@wh.iov.cn

homologs are extensively present in parasitoid wasps, alphabaculo-viruses, and lepidopteran insects but not in others (Fig. 1a). Some baculoviruses harbor two *p26* copies: copy-1 with and copy-2 without signal peptide. Due to the low sequence similarity between these two copies, it was hypothesized that they might have been acquired in independent events during evolution[9].

Melanization is a unique and prominent innate immune defense mechanism in the hemolymph of insects and crustaceans. Sequential

activation of an extracellular serine protease cascade converts pro-phenoloxidase (PPO) to active phenoloxidase (PO) to catalyze the formation of melanin[10–12]. This immune response is crucial for insects to combat viruses and other invading pathogens and, in turn, viruses have evolved versatile strategies to inhibit host melanization response to ensure their survival[13,14]. For example, we previously demonstrated that the baculovirus HearNPV successfully attenuates the melanization response of the cotton bollworm[15]. However, identity of the

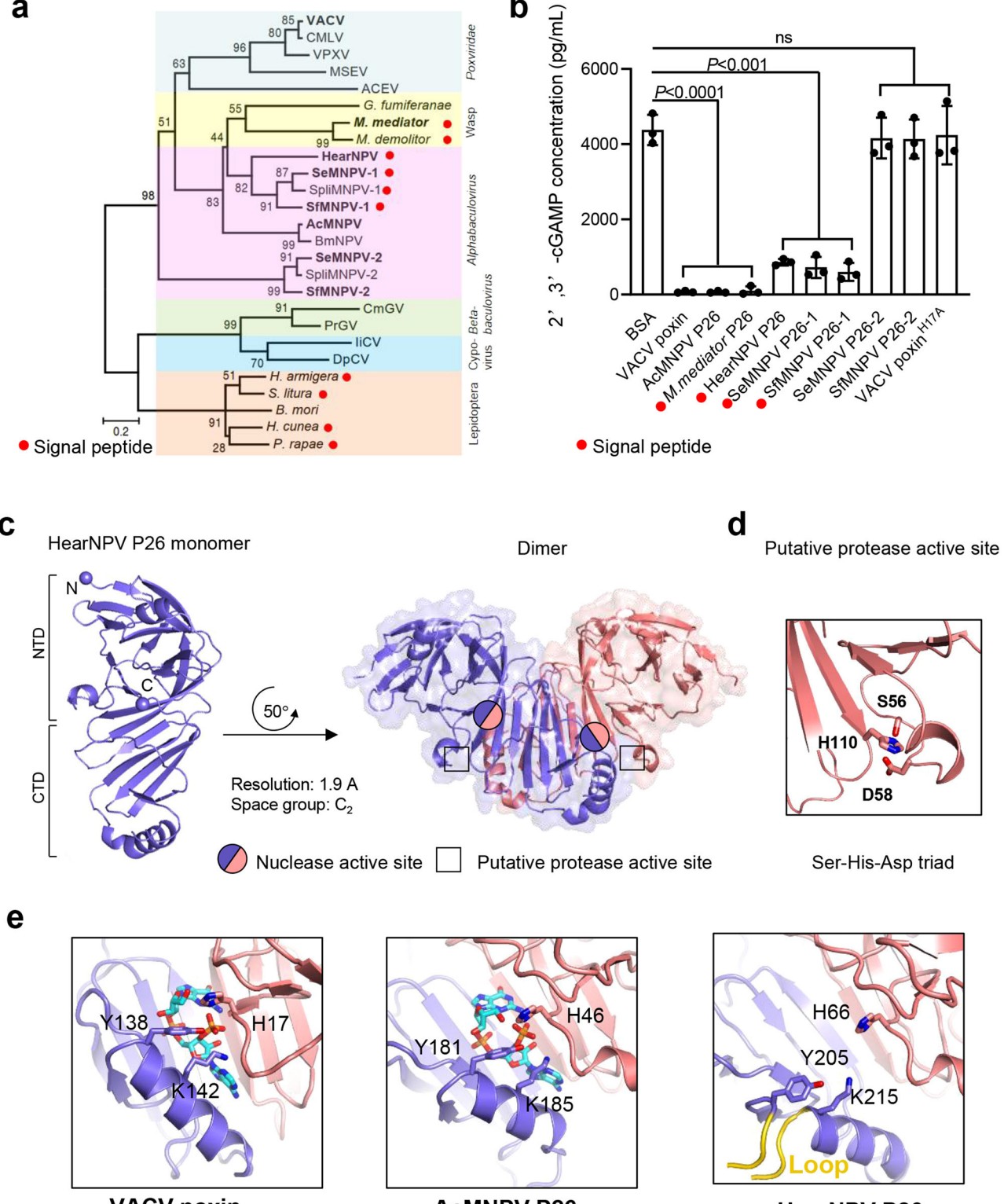

**a** Phylogenetic tree. Red dots indicate signal peptide.

**b** 2',3'-cGAMP concentration (pg/mL) bar chart comparing BSA, VACV poxin, AcMNPV P26, M. mediator P26, HearNPV P26, SeMNPV P26-1, SfMNPV P26-1, SeMNPV P26-2, SfMNPV P26-2, VACV poxin[H17A]. P<0.0001, P<0.001, ns. ● Signal peptide

**c** HearNPV P26 monomer (NTD, CTD). Dimer. 50°. Resolution: 1.9 Å. Space group: C₂. Nuclease active site. Putative protease active site.

**d** Putative protease active site. S56, H110, D58. Ser-His-Asp triad.

**e** VACV poxin (Y138, H17, K142). AcMNPV P26 (Y181, H46, K185). HearNPV P26 (H66, Y205, K215, Loop).

**Fig. 1 | Comparison of 2'3'-cGAMP nuclease activities and the active sites of different sources. a** Phylogenetic relationship of poxin/P26 homologs from poxviruses, parasitoid wasps, alphabaculoviruses, betabaculoviruses, cypoviruses, and lepidopteran insects. Numbers on the nodes indicate the maximum-likelihood (ML) tree nonparametric bootstrap supports over 50 (1000 replicates). **b** ELISA analysis of 2'3'-cGAMP degradation activities of selected poxin/P26 proteins (names in bold in **a**). Values are expressed as mean ± s.e.m of three independent experiments. Statistical significance of differences was evaluated via two-tailed student's $t$ test. BSA vs. VACV poxin: $P < 0.0001$, BSA vs. AcMNPV P26: $P < 0.0001$, BSA vs. *M. mediaror* P26: $P < 0.0001$, BSA vs. HearNPV P26: $P < 0.001$, BSA vs. SeMNPV P26-1: $P < 0.001$, BSA vs. SfMNPV P26-1: $P < 0.001$, BSA vs. SeMNPV P26−2: $P = 0.61$, BSA vs. SfMNPV P26-2: $P = 0.56$, BSA vs. VACV poxin[H17A]: $P = 0.80$, ns not significant. The red spot represents signal peptide in (**a**, **b**). **c** Crystal structure of HearNPV P26 (PDB code: 7WN7). The monomer of P26 is represented in ribbon model (left) with the N-terminal domain (NTD) and C-terminal domain (CTD) labeled. The dimer of P26 is shown as semi-transparent surface and cartoon representations (right). The two subunits are colored blue and pink, and the nuclease active site and the putative protease active site are labeled with a circle and rectangle, respectively. **d** The potential serine protease active site of HearNPV P26. The serine protease-like catalytic triad (S-H-D) is shown as sticks. **e** Structural comparison of the active site of 2'3'-cGAMP nucleases of VACV poxin (PDB code: 6EA9), AcMNPV P26 (PDB code: 6XB3) and HearNPV P26 (PDB code: 7WN7). The nuclease active-site residues (H-Y-K) and the bound substrate (2'3'-cGAMP) are shown as sticks. The additional 9-residue loop between the Y-K residues is highlighted yellow in HearNPV P26. Source data are provided as a Source Data file.

### Table 1 | Information of baculovirus and wasp P26 homologs

| Abbreviation | Accession | Full name | Signal peptide | Typical insect hosts |
|---|---|---|---|---|
| AcMNPV P26 | NP_054166.1 | Autographa californica nucleopolyhedrovirus hypothetical protein ACNVgp137 | No | *Spodoptera frugiperda; Trichoplusia ni* |
| *M.mediator* P26 | UXE46284.1 | *Microplitis mediator* P26 | Yes | *Helicoverpa armigera; Mythimna separata* |
| HearNPV P26 | NP_075091.1 | Helicoverpa armigera nucleopolyhedrovirus G4 ORF22 | Yes | *Helicoverpa armigera* |
| SeMNPV P26-1 | NP_037889.1 | Spodoptera exigua multiple nucleopolyhedrovirus ORF129 | Yes | *Spodoptera exigua* |
| SeMNPV P26-2 | NP_037847.1 | Spodoptera exigua multiple nucleopolyhedrovirus ORF87 | No | *Spodoptera exigua* |
| SfMNPV P26-1 | YP_001036423.1 | Spodoptera frugiperda multiple nucleopolyhedrovirus ORF131 | Yes | *Spodoptera frugiperda* |
| SfMNPV P26-2 | YP_001036378.1 | Spodoptera frugiperda multiple nucleopolyhedrovirus ORF86 | No | *Spodoptera frugiperda* |

NP_054166.1
NP_075091.1
NP_037889.1
NP_037847.1
YP_001036423.1
YP_001036378.1
UXE46284.1

baculoviral gene(s) responsible for suppressing this extracellular innate immune response largely remains unknown.

Here, we provided first evidence to show that the baculovirus HearNPV P26 inhibits host PPO activation by targeting a key cofactor, serine protease homologs (SPHs), of PPO activating protease[11]. We then determined the crystal structure of HearNPV P26 and performed structure-based mutagenesis and functional analysis to identify sites crucial for its functions. Finally, we confirmed a broad-spectrum melanization-inhibitory effect of the secreted P26s from different baculovirus and wasp origins.

## Results

### Nuclease activity of P26 homologs is not correlated with protein secretion

Since not all P26 homologs contain predicted nuclease active-site residues (H-Y-K) and signal peptide (Supplementary Fig. 1), we first asked whether the cGAMP nuclease activity is a universal feature of P26s and whether the activity is associated with secretion. Several representative baculoviral P26 homologs along with *M. mediator* P26 and vaccinia virus (VACV) poxin were expressed in bacteria and purified for the 2'3'-cGAMP nuclease activity assay (Table 1 and Supplementary Fig. 2a). The results showed that VACV poxin (no signal peptide), AcMNPV P26 (no signal peptide), and wasp *M. mediator* P26 (with signal peptide) had the highest 2'3'-cGAMP nuclease activity; the three signal peptide-containing baculoviral P26 proteins showed moderate activities of the nuclease, while the two baculoviral copy-2 P26s (no signal peptide) and the negative control VACV poxin[H17A] mutant lacked nuclease activity (Fig. 1b). Therefore, the nuclease activity is neither conserved in all P26 homologs nor associated with protein secretion, which is consistent with results from a recent study[2].

### The crystal structure of baculovirus HearNPV P26

To provide structural insights into the activity differences in 2'3'-cGAMP degradation across P26 homologs, we determined the crystal structure of P26 from HearNPV, a representative baculovirus with a single copy of secreted P26, at a resolution of 1.9 Å (Supplementary Table 1). Similar to other reported poxin/P26 structures[1], HearNPV P26 contains an N-terminal β-barrel domain and a C-terminal β-sandwich domain, and assembles into a V-shaped dimeric complex in the crystal lattice, with a putative serine protease-like active-site (S56-H110-D58) located in the N-terminal domain of each monomer and the putative nuclease active sites (H66-Y205-K215) at the interface of the two monomers (Fig. 1c, d). Similar to previously determined VACV poxin and AcMNPV P26 structures, a conserved H-Y-K catalytic triad for 2'3'-cGAMP hydrolysis is located in a deep cleft of the HearNPV P26 structure at the dimer interface between the N-terminal domain from one monomer and the C-terminal domain from the other. A loop substructure is present between Y205 and K215 in the HearNPV P26 structure, while these two residues are located in a single helix in the structures of VACV poxin and AcMNPV P26 (Fig. 1e). Structural modeling indicated that both SeMNPV P26-1 and SfMNPV P26-1 with moderate nuclease activity also have an inserted loop between the Y and K residues, whilst the *M. mediator* P26 with high nuclease activity does not, implying that the poxin/P26 nuclease activity could be modulated by this insertion. The two baculoviral copy-2 P26s lack the nuclease activity, likely due to the absence of H-Y-K catalytic center (Fig. 1b and Supplementary Fig. 3).

### HearNPV P26 inhibits host melanization response

Led by these pieces of evidence, we examined whether P26 is involved in the extracellular melanization pathway, using HearNPV and cotton

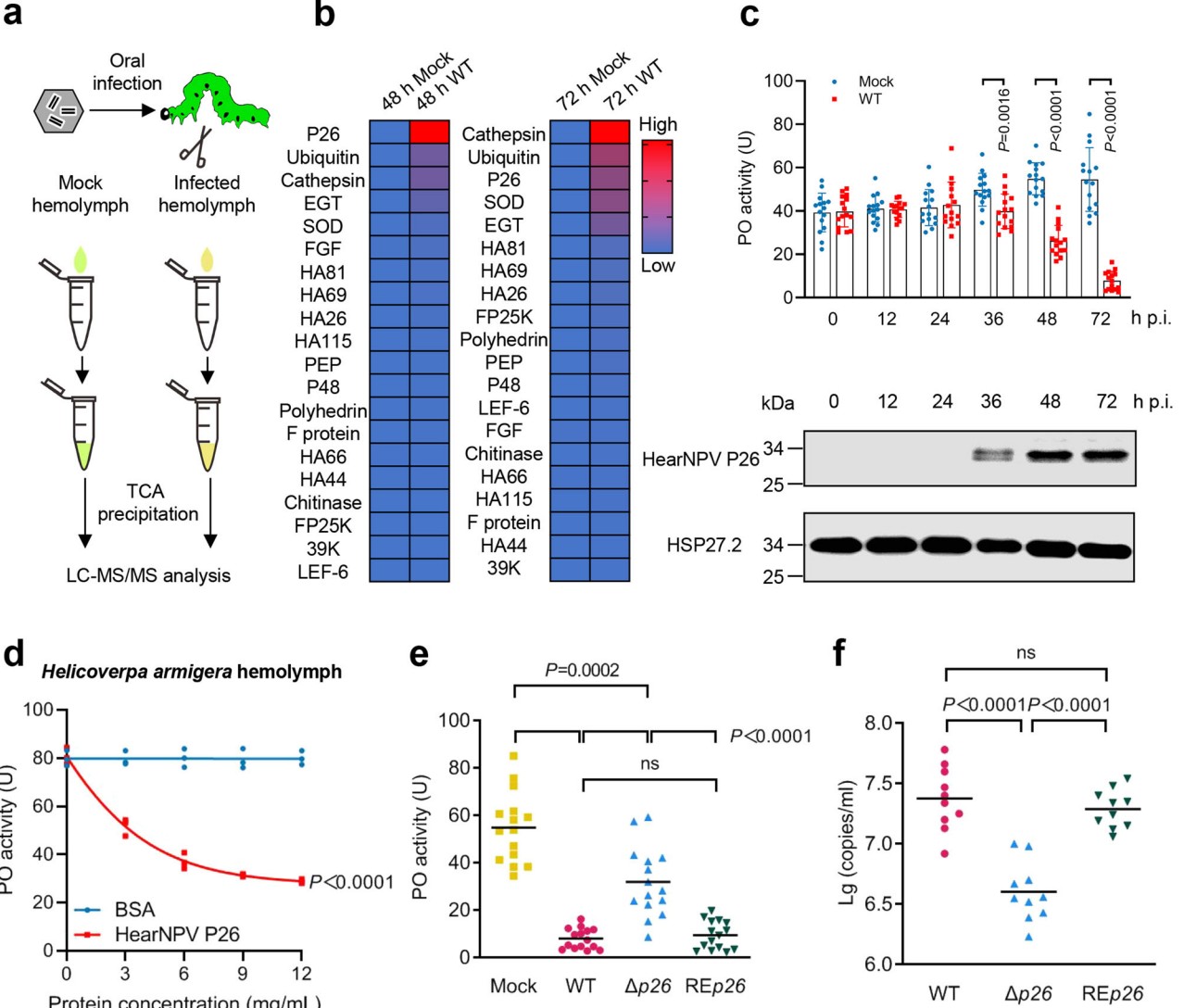

**Fig. 2 | HearNPV P26 inhibits the prophenoloxidase (PPO) activation. a** Scheme of mass spectrometry of hemolymph proteins from HearNPV-infected and mock-infected *H. armigera* larvae. **b** Heatmap of differentially expressed viral proteins in the hemolymph at 48 and 72 h p.i.. Values are expressed as mean ± s.e.m of two independent experiments. **c** Time-course analyses of PO activity and P26 expression in the hemolymph of *H. armigera*. The dots represent individual *H. armigera*, and the bars indicate the mean ± s.e.m values of each group (*n* = 15). Statistical significance of differences between Mock-infected and WT HearNPV-infected was evaluated via two-tailed student's *t* test, 0 h p.i.: *P* = 0.89, 12 h p.i.: *P* = 0.88, 24 h p.i.: *P* = 0.73, 36 h p.i.: *P* = 0.0016, 48 h p.i.: *P* < 0.0001, 72 h p.i.: *P* < 0.0001. Upper panel: comparison of PO activities in hemolymph samples from mock-infected and virus-infected *H. armigera* at the indicated time points. Lower panel: Western blot analysis of the P26 expression profile in the hemolymph collected at the same time points post infection. HSP27.2[15] was used as loading control. **d** Concentration-dependent inhibition of hemolymph melanization by P26 proteins in vitro. Hemolymph (1 μL) from naïve third instar larvae, mixed with different amounts of P26 (0–12 mg/mL), was incubated on ice for 40 min before measuring PO activity.

BSA was used as a negative control. All data are representative of three independent experiments. Statistical significance of differences was evaluated via two-way ANOVA, BSA vs. HearNPV P26 *P* < 0.0001. **e** Comparison of PO activity in hemolymph from mock-infected, WT HearNPV-infected, Δ*p26* HearNPV-infected, and RE*p26* HearNPV-infected *H. armigera* at 72 h p.i.. The dots represent individual *H. armigera*, and the bars indicate the mean ± s.e.m values of each group (*n* = 15). Statistical significance of differences was evaluated via two-tailed student's *t* test, mock vs. WT: *P* < 0.0001, mock vs. Δ*p26*: *P* = 0.0002, WT vs. Δ*p26*: *P* < 0.0001, Δ*p26* vs. RE*p26*: *P* < 0.0001, WT vs. RE*p26*: *P* = 0.46. **f** Viral loads in the hemolymph. *H. armigera* larvae were infected with WT or Δ*p26* HearNPV. Number of DNA copies of Δ*p26* HearNPV in the hemolymph decreased by more than tenfold compared to WT and RE*p26*. The dots represent individual *H. armigera*, and the bars indicate the mean ± s.e.m values of each group (*n* = 10). Statistical significance of differences was evaluated via two-tailed Student's *t* tests, WT vs. Δ*p26*: *P* < 0.0001, Δ*p26* vs. RE*p26*: *P* < 0.0001, WT vs. RE*p26*: *P* = 0.38. ns not significant. Source data are provided as a Source Data file.

bollworm as a testing system. We first detected the presence of HearNPV P26 in hemolymph from the virus-infected insect via liquid chromatography tandem mass spectrometry (LC-MS/MS) analysis (Fig. 2a). P26 was one of the most abundant viral proteins detected in the infected hemolymph at 48 and 72 h post infection (h p.i.) (Fig. 2b and Supplementary Table 2), implying a role in the host hemolymph. We detected significantly decreased activity of hemolymph PO, a key enzyme of the melanization pathway, at 48 and 72 h post HearNPV

infection. Consistently, the P26 protein reached peak levels at these times (Fig. 2c). Therefore, we suggest that P26 accumulation in the hemolymph might be involved in viral counteraction against the melanization response. To test this premise, we expressed HearNPV P26 proteins using a *Drosophila* Expression System (Invitrogen) (Supplementary Fig. 2b) and found that purified P26 proteins reduced hemolymph PO activity in vitro in a dose-dependent manner, with PO activity decreasing by up to 64% at a protein concentration of

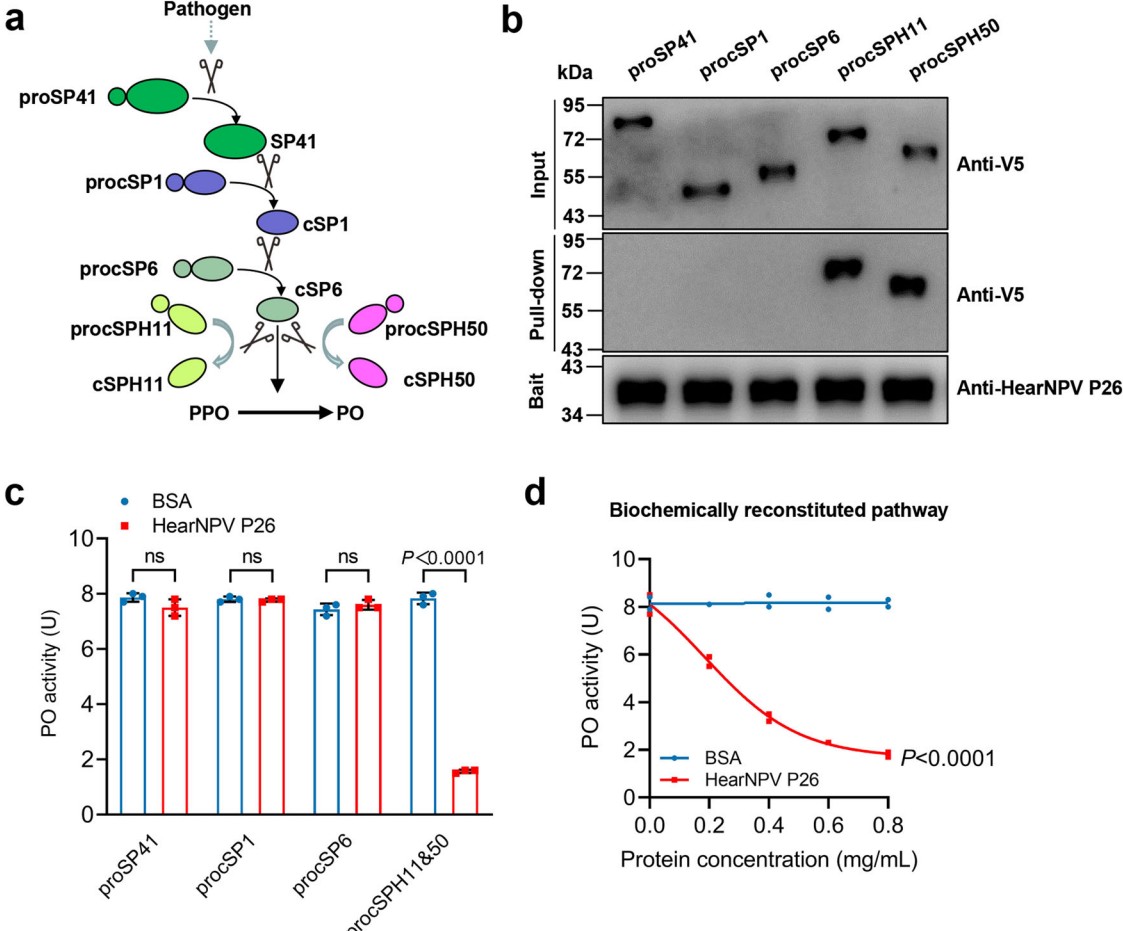

**Fig. 3 | Molecular mechanism for inhibiting PPO activation by HearNPV P26.**
**a** Scheme of the PPO activation pathway of *H. armigera* hemolymph. Activation of proSP41 initiates activation of downstream procSP1 and procSP6 successively, causing PPO activation. The cofactor precursors (procSPH11 and procSPH50) are activated by cSP6 to enhance PPO activation by cSP6. **b** Pull-down assay of HearNPV P26 with proSP41, procSP1, procSP6, procSPH11, and procSPH50. The recombinant HearNPV P26 contains a twin-strep tag, and proSP41, procSP1, procSP6, procSPH11, or procSPH50 has a V5 tag. The samples were analyzed by SDS-PAGE and immunoblotting using anti-V5 and anti-P26 antibodies. **c** Identification of the P26 targets in the melanization pathway. P26 was pre-incubated with proSP41$_{Xa}$, procSP1, procSP6, or procSPH11 and procSPH50, followed by incubation with the remaining components of the pathway prior to PO activity measurement. BSA was used as a negative control. Values are expressed as mean ± s.e.m of three independent experiments. Statistical significance of differences between BSA and HearNPV P26 was evaluated via two-tailed Student's *t* tests, proSP41: *P* = 0.13, procSP1: *P* = 0.64, procSP6: *P* = 0.35, procSPH11 and procSPH50: *P* < 0.0001. **d** Concentration-dependent inhibition of PPO activation by HearNPV P26. ProcSPH11 and procSPH50 were mixed with different concentrations of P26 (0−0.8 mg/mL), followed by addition of other components of the melanization pathway, before PO activity assay. BSA was used as a negative control. All data are representative of three independent experiments. Statistical significance of differences was evaluated via two-way ANOVA, BSA vs. HearNPV P26: *P* < 0.0001. ns not significant. Source data are provided as a Source Data file.

12 mg/mL (Fig. 2d, *P* < 0.0001). To determine the impact of P26 on PPO activation and viral infectivity in vivo, a *p26*-deleted (Δ*p26*) and a repaired (RE*p26*) HearNPV were constructed. *Helicoverpa armigera* larvae were infected with Δ*p26*, RE*p26*, and wild type (WT) viruses, and the PO activity and viral loads in the hemolymph were measured at 72 h p.i.. The results showed that significantly higher PO activity was detected in the infected hemolymph of Δ*p26* viruses than that of the hemolymph infected with WT and RE*p26* viruses (Fig. 2e). The DNA copy numbers of HearNPV in Δ*p26* infected hemolymph decreased to approximately 20% of the WT levels (Fig. 2f). Moreover, deletion of *p26* significantly extended the median survival time (ST$_{50}$) of the virus in infected larvae (Supplementary Table 3). The in vitro and in vivo data above suggest that HearNPV P26 functions as an inhibitor of the host melanization response, which is crucial for viral infection in vivo.

### HearNPV P26 targets the cofactor of PPO activation reaction
The PPO activation pathway of *H. armigera* is mediated by a cascade of three serine proteases SP41-cSP1-cSP6, and a combined cofactor of cSPH11 and cSPH50 (Fig. 3a)[16]. Upon activation by pathogen invasion, these components are converted from zymogens to active forms and finally activate PPO to catalyze melanin formation. Note that cSPH11 and cSPH50 act synergistically as a cofactor in PPO activation, and cSPH11 or cSPH50 alone does not enhance the activation reaction[16]. To investigate the molecular mechanism of P26-mediated inhibition of PPO activation, we first examined a possible interaction between P26 and component(s) in the cascade. The pull-down experiments showed that none of the serine proteases (proSP41, procSP1, or procSP6) interacted with P26, whereas procSPH11 and procSPH50 of the cofactor showed strong interactions with P26 (Fig. 3b). We then used an established in vitro PO activity assay[16] to further screen for P26 targets by pre-incubating it with different cascade component(s). The results showed that pre-incubation of P26 with proSP41, procSP1, or procSP6 did not affect the PPO activation; in contrast, when P26 was pre-incubated with procSPH11 and procSPH50, the PO activity was significantly decreased (Fig. 3c). Further, P26 could inhibit the cofactor-mediated PPO activation in a dose-dependent manner (Fig. 3d,

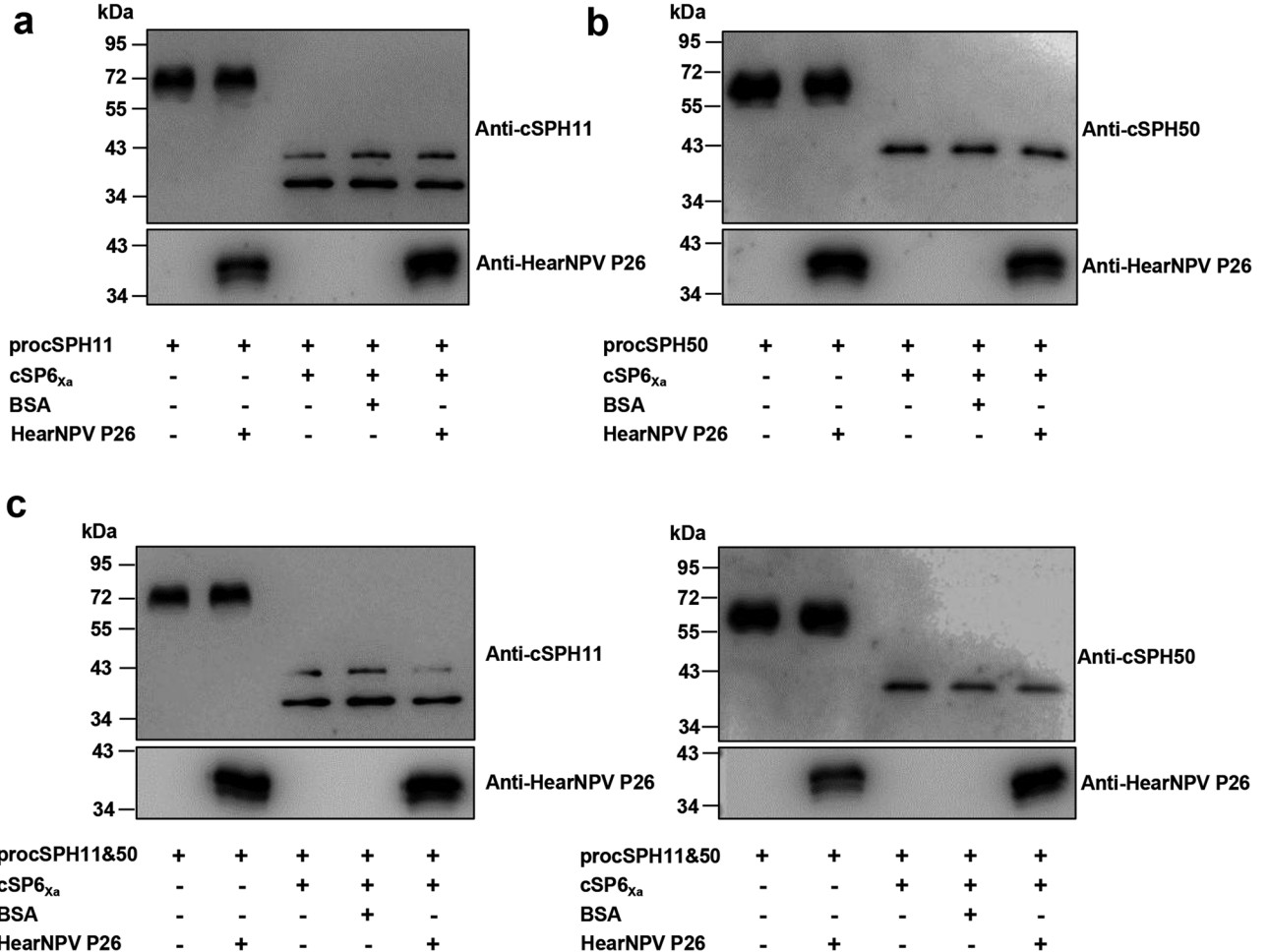

**Fig. 4 | The impact of HearNPV P26 on the proteolytic cleavages of the serine protease homologs.** cSP6$_{Xa}$ was activated by Factor Xa before incubating with procSPH11 (**a**), procSPH50 (**b**), or procSPH11 and procSPH50 (**c**) for 1 h. The mixtures were analyzed by immunoblotting using cSPH11, cSPH50, and HearNPV P26 antibodies. BSA was used as a negative control. Source data are provided as a Source Data file.

$P < 0.0001$). As procSPH11 and procSPH50 are cleaved by the upstream cSP6 during activation[16], we investigated whether interaction with P26 would affect their cleavage. Immunoblotting showed that pre-incubation of P26 with procSPH11, procSPH50, or procSPH11 and procSPH50 did not affect their cleavage by cSP6 (Fig. 4a–c).

### The important functional sites in HearNPV P26

To better understand the structure-function relationship of HearNPV P26, we focused on three characteristic regions identified by structural analysis: the putative serine protease-like catalytic triad (S56-H110-D58) (Fig. 1d), the nuclease active-site (H66-Y205-K215) (Fig. 1e), and the distinct electrostatic surface with negative or positive potentials (Fig. 5a). Three sets of mutations at the key residues within these regions were designed and expressed using the *Drosophila* S2 Cell Expression System to determine their functions in terms of suppressing PPO activation (Supplementary Fig. 2c). The point mutations that changed the surface charge of the protein (D241H/E243H, H66D/H67D, D241H/E243H/H66D/H67D) had no impact on the suppression of PPO activation (Fig. 5b). Notably, two mutants harboring a combination of mutations in surface charge residues also contained the key H66 residue for nuclease activity, suggesting that the nuclease activity of P26 is not correlated with its function of suppressing PPO activation. Mutation of the key nuclease activity residue H66 (H66A) led to complete abrogation of cGAMP hydrolysis activity (Fig. 5d). However, this mutant did not show a compromised ability to inhibit PPO activation (Fig. 5e), further confirming that the nuclease activity is independent of melanization inhibition. In contrast, simultaneous mutations of the three key residues (S56A/H110A/D58A) constituting the putative serine protease catalytic triad in P26 completely abolished the suppression of PPO activation ($P < 0.0001$); the H110A ($P < 0.0001$) and S56A ($P < 0.05$) point mutation also significantly impaired P26 function, while the D58A mutant had no significant influence (Fig. 5c). Notably, both S56A/H110A/D58A and H110A showed levels of cGAMP nuclease activity comparable to the WT P26 (Fig. 5d). The results also showed that the melanization-inhibitory function of the P26 is independent of its nuclease activity.

### Broad-spectrum melanization-inhibitory function of signal peptide-containing P26s/poxins

To investigate whether the extracellular role of antagonizing host melanization is evolutionarily conserved among diverse P26/poxin homologs, the eight P26/poxin proteins tested earlier in Fig. 1b (except poxin H117A mutant) were expressed in the *Drosophila* S2 Cell Expression System and further purified (Supplementary Fig. 2b) to test their inhibitory activity on PPO activation. The results showed that all the signal peptide-containing P26s, including those from baculoviruses and a parasitoid wasp, could inhibit hemolymph PPO activation of *H. armigera*, with the highest inhibitory effect achieved by HearNPV P26. In contrast, the ones without signal peptide did not possess this ability (Fig. 6a). To further confirm this finding, the experiment was conducted using hemolymph from another lepidopteran insect *Spodoptera frugiperda*. As in *H. armigera*, only the signal peptide-

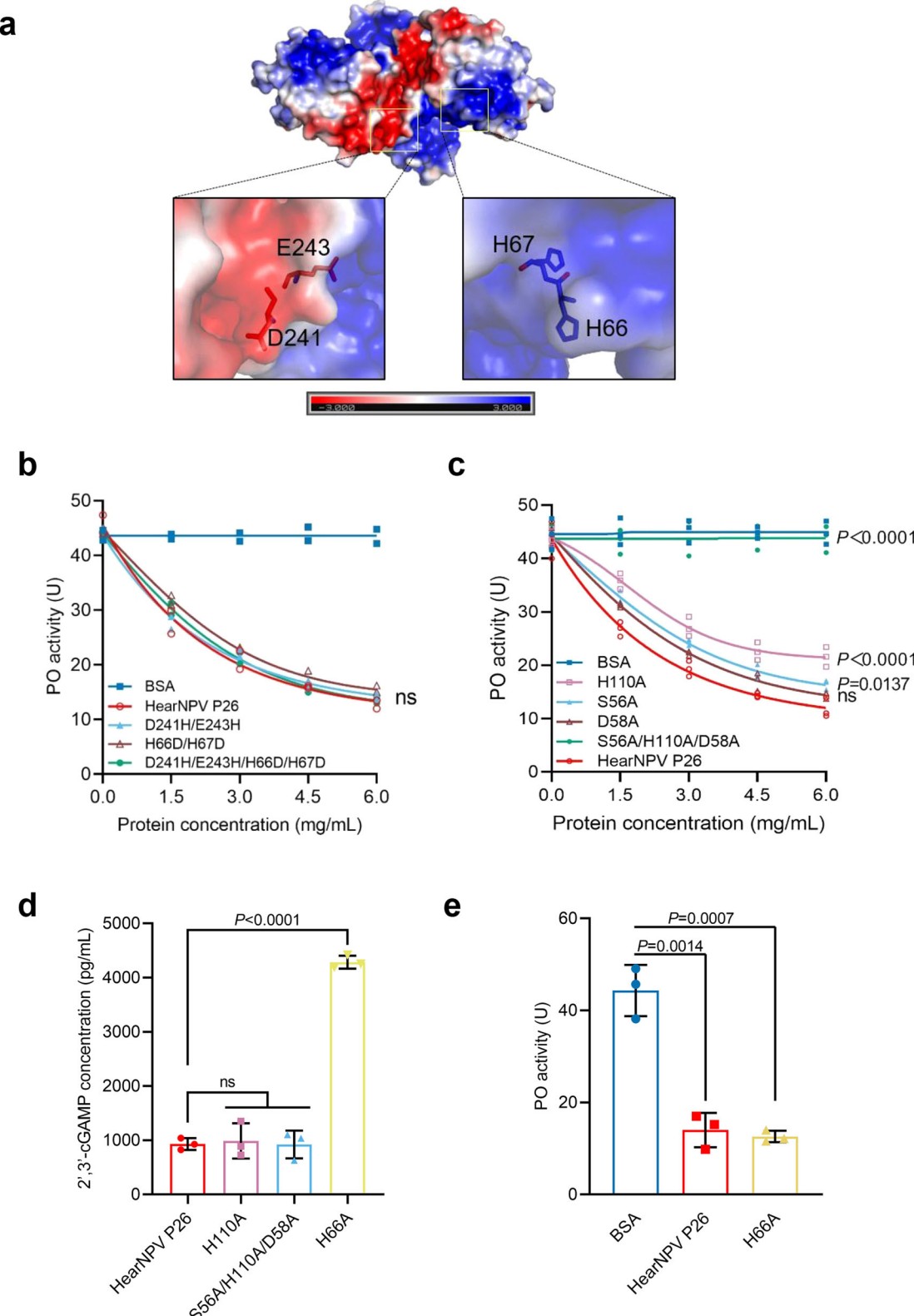

containing P26s were able to inhibit the hemolymph melanization reaction of *S. frugiperda*, with the highest inhibitory effect achieved by SfMNPV P26 copy-1, a protein from the native virus to which the insect is susceptible (Fig. 6b). These findings suggest that the secreted P26 homologs of baculoviruses and parasitoids can suppress extracellular immune function and seem to have evolved adaptively with the host immune system.

## Discussion

In summary, we have unraveled a novel extracellular function of P26/poxin, thereby demonstrating a single protein capable of suppressing both intracellular and extracellular immune responses (Fig. 7). The viral P26 is an antagonist of host melanization, targeting the cofactor downstream of the PPO activation pathway, at least in *H. armigera*. The ability to inhibit the extracellular response is uniquely associated with

**Fig. 5 | Mutational analysis of key residues in HearNPV P26. a** Electrostatic surface representation of HearNPV P26 dimer. The positive and negative electrostatic potentials are shown in blue and red, respectively. The conserved acidic and basic residues are shown as sticks. Scale bar: −3 kT/e in red to +3 kT/e in blue. **b** Concentration-dependent inhibition of PPO activation by HearNPV P26 proteins mutated in surface charged residues. The hemolymph (1 μL) from naïve third instar larvae was mixed individually with 0–6 mg/mL HearNPV P26 mutant proteins and incubated on ice for 30 min before PO activity measurement. BSA was used as a negative control. All data are representative of two independent experiments. Statistical significance of differences between HearNPV P26 and mutants were evaluated via two-way ANOVA, P26 vs. D241H/E243H: $P = 0.40$, P26 vs. H66D/H67D: $P = 0.11$, P26 vs. D241H/E243H/H66D/H67D: $P = 0.18$. **c** Concentration-dependent inhibition of PPO activation by 0–6 mg/mL HearNPV P26 proteins mutated at the serine protease-like catalytic triad. BSA was used as a negative control. All data are

representative of three independent experiments. Statistical significance of differences between HearNPV P26 and mutants were evaluated via two-way ANOVA, P26 vs S56A/H110A/D58A: $P < 0.0001$, P26 vs. H110A: $P < 0.0001$, P26 vs. S56A: $P = 0.014$, P26 vs. D58A: $P = 0.28$. **d** 2′3′-cGAMP nuclease activity analysis of the putative P26 protease active-site mutants (H110A, S56A/H110A/D58A) and the 2′3′-cGAMP nuclease activity mutant (H66A). Values are expressed as mean ± s.e.m of three independent experiments. Statistical significance of differences was evaluated via two-tailed Student's t tests. P26 vs. H110A: $P = 0.78$, P26 vs. S56A/H110A/D58A: $P = 0.95$, P26 vs. H66A: $P < 0.0001$. **e** PO activity analysis of the P26 mutant (H66A) for 2′3′-cGAMP nuclease activity. Values are expressed as mean ± s.e.m of three independent experiments and bars indicate the mean values of each group. Statistical significance of differences was evaluated via two-tailed Student's t test. BSA vs. P26: $P = 0.0014$, BSA vs. H66A: $P = 0.0007$. ns not significant. Source data are provided as a Source Data file.

the signal peptide-containing P26s. The wide distribution of secreted P26/poxins suggests that the suppression of PPO activation is an ancient mechanism recruited by parasitoids and insect viruses. The ability to inhibit melanization and the nuclease function are structurally and functionally independent. Among the P26/poxin proteins tested, the signal peptide-containing ones appear to contain both extracellular and intracellular immune suppression functions, whilst the ones without signal peptide either only harbor intracellular cGAMP degrading activity or lack both extracellular and intracellular activities, suggesting the occurrence of gain- and loss-of-function events during P26 evolution.

The results of this study raised some new questions. Here, we have not investigated the function of P26/poxin homologs in lepidopteran insects, which include proteins both with and without signal peptide. Some of the lepidopteran P26/poxin showed nuclease function[1,2], thus it would be interesting to investigate their function as PPO activation inhibitors. Previous proteomic and transcriptomic data showed that *H. armigera* P26 (signal peptide-containing) is one of the differentially expressed genes after inoculation of a wide variety of pathogens, including fungi, baculovirus and wasp (NCBI SRA: PRJNA349107, https://www.ncbi.nlm.nih.gov/bioproject/PRJNA349107/)[17,18], indicating it is an immune responsive gene. Its possible role in regulating the insect immune system deserves further investigation, as it would elucidate how P26/poxin genes evolved during long-term adaption between insect hosts and pathogens.

In mammals, the intracellular cGAS-STING pathway plays a pivotal role in antiviral defense. cGAS produces 2′3′-cGAMP, which can be degraded by virus-encoded poxins[1]. In lepidopteran insect *Bombyx mori*, baculovirus BmNPV (a dsDNA virus) induces host cellular 2′3′-cGAMP production at an early stage of infection, which activates STING-mediated antiviral signaling pathway, although its upstream receptor is still unclear[19]. Additionally, BmNPV-encoded P26 can efficiently degrade 2′3′-cGAMP[1]. Interestingly, two cGAS-like receptors (cGLRs), cGLR1 and cGLR2, were recently identified in *Drosophila*[20,21]. Like cGAS in mammals, they synthesize cyclicdinucleotides to activate STING. However, *Drosophila* cGLR1 senses dsRNA ligand instead of DNA ligand to produce a novel cGAMP isomer 3′2′-cGAMP, which shows higher potency against RNA viruses than 2′3′-cGAMP. And 3′2′-cGAMP has been shown to be resistant to poxin cleavage[20]. Therefore, it will be interesting to investigate if insect RNA viruses encode any nuclease against this signaling pathway (Fig. 7).

Besides its intracellular role, 2′3′-cGAMP also exists as a soluble, extracellular immunotransmitter[22–24]. Therefore, it is possible that apart from being the melanization inhibitor in insects, the secretory P26/poxin homologs might also be involved in degrading extracellular 2′3′-cGAMP in mammals and insects (Fig. 7). These hypotheses, of course, need further study.

The detailed mechanism of how P26 suppresses PPO activation is unclear. Although HearNPV P26 contains a serine protease-like triad, but the residues do not occur in the typical order of H-D-S in

the primary structure. And the serine protease-like triad is present in only some P26 homologs (Supplementary Fig. 4). We tried to overlay the structure of HearNPV P26 with that of Zika virus NS3 protease, as well as with that of lepidopteran *Trichoplusia ni* poxin. The results showed that the protease-like triad of HearNPV P26 does not overlap with the other two (Supplementary Fig. 5). In addition, the mutation of the putative catalytic serine residue S56 showed only minor impact on suppression of PO activity (Fig. 5c). Most importantly, we failed to show proteolytic cleavage activity (Fig. 4). Thus, it is unlikely that HearNPV P26 acts through a functional protease domain. Moreover, the detailed mechanism of the cofactor in the PPO activation reaction remains unclear. The cofactor is composed of two noncatalytic SPHs which contain a clip domain at their amino terminus (cSPHs)[25]. PPO activation occurs efficiently in the presence of the cofactor, and the cSPHs are covalently associated with PO to form a high $M_r$ melanization complex[25]. According to the most thoroughly studied *Manduca sexta* model, two components of the cofactor form scaffold structures to orient PPO activating protease and PPO for optimal cleavage[26]. Therefore, P26 may bind to cSPHs and physically block the scaffolds and consequently inhibit the activation of PPO. Since the PPO activation pathway usually contains more than one upstream branch, targeting downstream components appears to be an effective strategy and has been reported in other invertebrate viruses, such as polydnaviruses (carried by many parasitoid wasps) and white spot syndrome virus[27–29].

It remains unclear how ancestral P26/poxin evolved to perform dual functions. Eaglesham et al. showed that poxins were descended from self-cleaving RNA-virus proteases[2]. The distinct amarilloviral proteases they identified, however, lack the signal peptide for secretion, which is a prerequisite for melanization-inhibitory activity. Studies dedicated to addressing the above questions will further shed light on the function-evolution relationships and precise working mechanism of this crucial viral immune inhibitor.

## Methods

### Insect cells, insects, and viruses

The *Helicoverpa zea* cell line HzAM1 was maintained at 28 °C in Grace's insect medium (Sigma-Aldrich, G8142) supplemented with 10% fetal bovine serum (FBS) (Gibco, 10091148). The S2 cell line (Invitrogen, R69007) was maintained at 27 °C in ESF921 medium (Expression Systems, 96-001-01). *H. armigera* larvae were reared on an artificial diet at 27 °C. The control virus vHaBac-*egfp-ph* was previously constructed in our laboratory[30].

The *p26* gene from HearNPV bacmid HaBacHZ8 was knocked out and repaired using the method of homologous recombination, as previously described[31]. HzAm1 cells were transfected with the recombinant bacmids, using Lipofectin (Invitrogen, 18292037) according to the manufacturer's protocol. Five days post-transfection, the culture supernatant containing infectious viruses was collected and used to

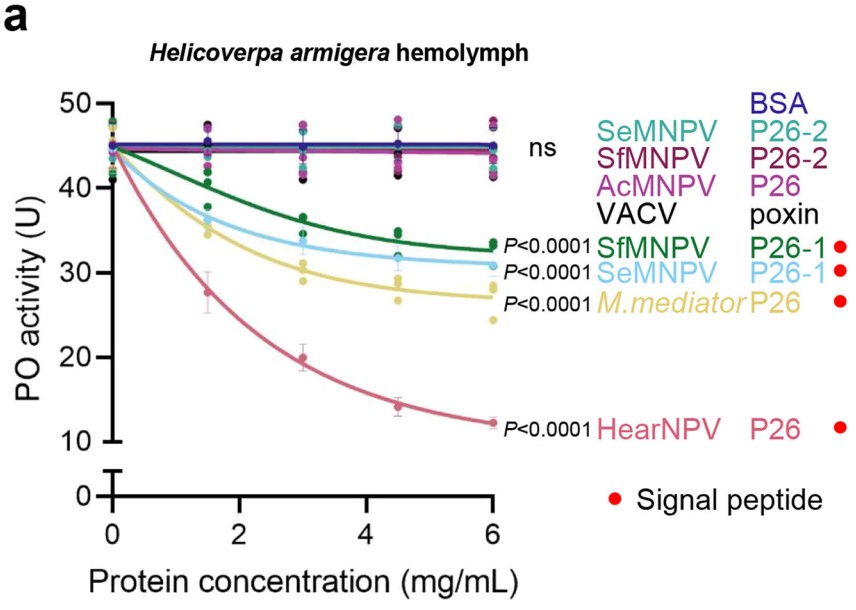

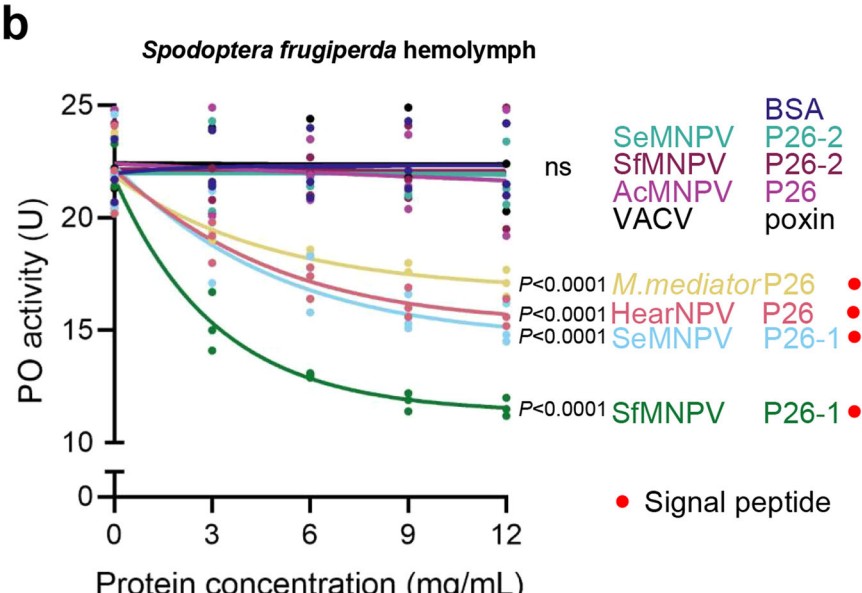

**Fig. 6 | Secreted P26 homologs from different sources evolved the function of suppressing PPO activation.** One µL hemolymph from naïve third instar *H. armigera* (**a**) or fourth instar *S. frugiperda* (**b**) were mixed with different amounts of indicated P26/poxin and incubated on ice for 30 min before PO activity assay. BSA was used as a negative control. All data are representative of three independent experiments. Statistical significance of differences between the negative control BSA and recombinant proteins were evaluated via two-way ANOVA. **a** BSA vs. SfMNPV P26-1: $P < 0.0001$, BSA vs. SeMNPV P26-1: $P < 0.0001$, BSA vs. *M. mediaror*

P26: $P < 0.0001$, BSA vs. HearNPV P26: $P < 0.0001$, BSA vs. AcMNPV: $P = 0.78$, BSA vs. SeMNPV P26-2: $P = 0.77$, BSA vs. SfMNPV P26-2: $P = 0.48$, BSA vs. VACV poxin: $P = 0.90$. **b** BSA vs. *M. mediaror* P26: $P < 0.0001$, BSA vs. HearNPV P26: $P < 0.0001$, BSA vs. SeMNPV P26-1: $P < 0.0001$, BSA vs SfMNPV P26-1: $P < 0.0001$, BSA vs. AcMNPV: $P = 0.22$, BSA vs. SeMNPV P26-2: $P = 0.46$, BSA vs. SfMNPV P26-2: $P = 0.92$, BSA vs. VACV poxin: $P = 0.45$. ns not significant. Source data are provided as a Source Data file.

infect HzAm1 cells to amplify the recombinant viruses vHacBacΔ*p26-egfp-ph* and vHacBacRe*p26-egfp-ph*.

**Prokaryotic expression of recombinant proteins**
The open reading frames of *p26/poxin* homologs including HearNPV *p26*, VACV *poxin*, VACV *poxin*[H17A], AcMNPV *p26*, SfMNPV *p26*–1, SfMNPV *p26*-2, SeMNPV *p26*–1, SeMNPV *p26*-2, and *M. mediaror p26* with signal peptide sequences deleted were amplified using viral genomic DNA as templates. PCR products were cloned into the **pET-28a(+)** (Novagen) or **pET-32a(+)** (Novagen) vector containing an N-terminal histidine tag (His-tag) and His-tag plus a thioredoxin tag (Trx-tag), respectively. Recombinant plasmids were transformed into

*E. coli* strain BL21 (DE3) for protein expression. The proteins were purified by HIS-Select Nickel Affinity Gel (Millipore, P6611) and Superdex-200 column (GE, 17-5175-01). For the HearNPV P26 samples used in crystallization, both the N- terminal His-tag and Trx-tag of the proteins were removed by thrombin digestion. The cleaved and purified proteins were concentrated to 10 to 20 mg/mL, then flash frozen by liquid nitrogen as single-use aliquots and stored at −80 °C.

Selenomethionine-labeled HearNPV P26 proteins were expressed in BL21 (DE3) cells using M9 minimal medium (47.5 mM $Na_2HPO_4$, 22 mM $K_2HPO_4$, 8.5 mM NaCl, 18.7 mM $NH_4Cl$, 0.4% w/v glucose, 2 mM $MgSO_4$, 0.1 mM $CaCl_2$) supplemented with 100 mg/L lysine, phenylalanine, and threonine; 50 mg/L isoleucine, leucine, and valine; and

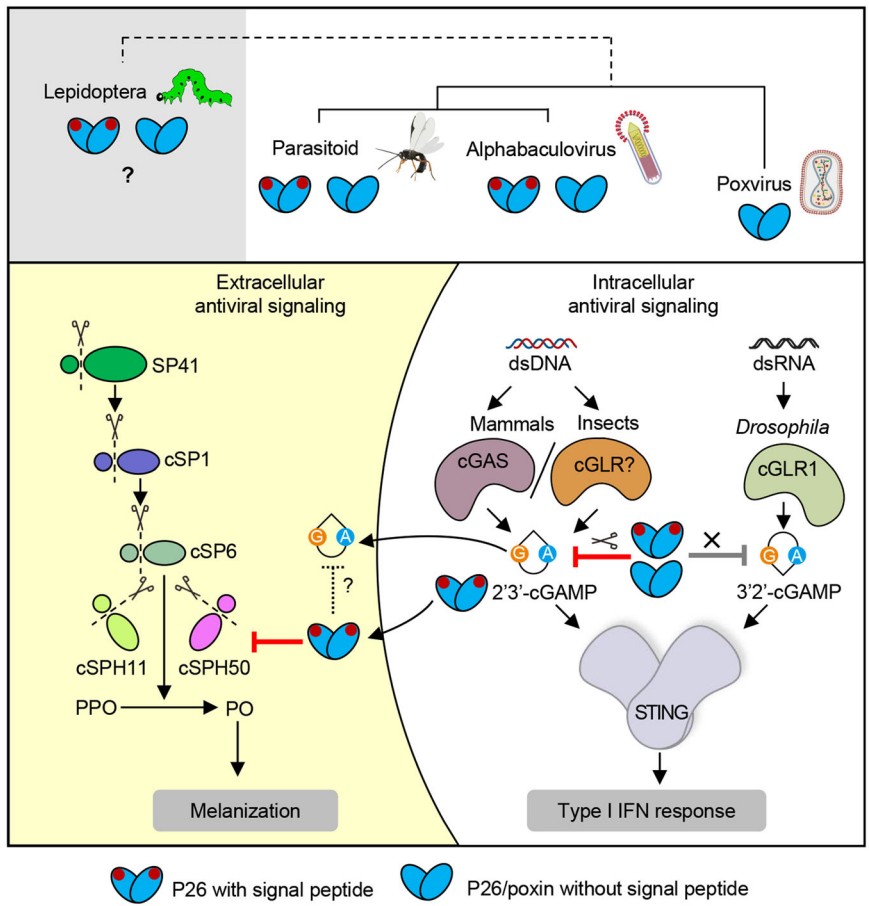

**Fig. 7 | A model depicting the extracellular and intracellular roles of immunosuppression of poxin/P26.** The signal peptide-containing P26s/poxins (labeled with red spot) from alphabaculoviruses or parasitoid wasps appear to contain both extracellular and intracellular immune suppression functions, whilst those without signal peptide either only harbor intracellular cGAMP degrading activity or lack both activities. Whether or not secreted P26 homologs from lepidopteran hosts function as extracellular melanization inhibitors is unclear and, thus, shown in gray background. In the cytoplasm of mammalian and insect cells, 2′

3′-cGAMP produced by cGAS or cGAS-like receptor (cGLR) in response to dsDNA ligand can be inhibited by poxin/P26's nuclease activity; while a novel 3′2′-cGAMP generated in *Drosophila* by cGLR1 in response to dsRNA ligand stimulation is resistant to poxin/P26. In insect hemolymph, secreted poxin/P26 proteins inhibit insect melanization pathway by targeting the key co-factor of a PPO activating protease, at least in baculovirus HearNPV-*H.armigera* infection system. They may also degrade extracellular 2′3′-cGAMP, which needs further study.

60 mg/L selenomethionine (SeMet, Sigma, 1611955). The protein was purified using the procedure described above.

### 2′3′-cGAMP nuclease activity assays

To determine 2′3′-cGAMP hydrolysis activity of recombinant proteins, 1 µL of each recombinant protein (2 µM) and 1 µL of 2′3′-cGAMP (21 µM) were added to 8 µL of buffer (50 mM HEPES-KOH pH 7.5, 40 mM KCl, 1 mM DTT). Reactions were carried out at 37 °C for 1 h. Then, each mixture was diluted to 3 mL and transferred to a 10 kDa ultrafiltration tube (Millipore, UFC8010), followed by centrifugation at 4000 g for 5 min at 4 °C. After centrifugation, 50 µL of each supernatant was used to assay the amount of 2′3′-cGAMP with the 2′3′-cGAMP ELISA Kit (Cayman, 501700), according to the manufacturer's instructions. Each test was performed three times. Results were analyzed statistically via two-tailed Student's t test.

### Protein crystallization, X-ray diffraction data collection, and structure determination

Crystallization screening was performed at 16 °C by vapor diffusion in sitting drops containing 0.4 µL of protein (10 mg/mL) and 0.4 µL of precipitant/reservoir solution. Native and SeMet-labeled HearNPV P26 crystals were obtained with a reservoir solution of 0.2 M ammonium sulfate, 0.1 M MES monohydrate (pH 6.7), and 28% (w/v) polyethylene

glycol monomethyl ether 5000. For collection of X-ray diffraction data, the crystals were harvested and transferred into the reservoir solution, followed by incremental replacement of a cryosolution (reservoir solution supplemented with 20% [vol/vol] glycerol) before flash frozen by liquid nitrogen.

Diffraction data were collected at BL18U1 and BL17U1 beamline at the Shanghai Synchrotron Radiation Facility (SSRF). For native HearNPV P26 crystals, 180 degrees of data was collected in 1° oscillation steps at 0.97780 Å. For SeMet-labeled crystals, 720 degrees of single-wavelength anomalous diffraction (SAD) data was collected in 1° oscillation steps at the Se-K edge (0.97915 Å). Diffraction data were integrated, merged, and scaled using the HKL2000 program[32]. The SeMet P26 structure (2.7 Å) was solved using the AutoSol[33] program in PHENIX Suite[34] by Se atom site search, phase calculations, density modification, and automated model-building. The native P26 structure (1.9 Å) was obtained via molecular replacement (MR) program PHASER[35] using the SeMet P26 structure as the initial search model. After several cycles of manual model rebuilding in Coot[36] and automated refinement in PHENIX[34], the final model was refined to have $R_{work}$ and $R_{free}$ values of 0.183 and 0.211, respectively. Structural superimpositions of HearNPV P26 with other correlated proteins were performed using the maximum likelihood-based structure superpositioning program THESEUS[37].

## Sample preparation for mass spectrometry

Cell-free hemolymph (100 µL) was collected from mock and recombinant virus-infected *H. armigera* larvae at 48 and 72 h p.i. with the addition of 1 mM N-phenylthiourea (Sigma, P7629) and 1 mM *p*-aminobenzamidine (Sigma, 857661). All samples were precipitated by adding 25 µL 100% w/v trichloroacetic acid (Sigma, T6399) and incubated on ice for 30 min. The protein pellet was collected by centrifugation at $140,000 \times g$ for 10 min at 4 °C, and the supernatant was removed. Cold acetone at −20 °C was used to wash the protein pellet, followed by centrifugation at $140,000 \times g$ for 10 min at 4 °C, after which acetone was removed. This step was repeated twice, and the remaining solvent was allowed to evaporate. Two technical replicates were performed for each sample. The dissolved protein pellet from each sample was reduced, alkylated, followed by digestion with trypsin. The purified peptides were subjected to LC-MS/MS and analyzed on a Q Exactive HF mass spectrometer coupled with an UltiMate 3000 RSLCnano system (Thermo Fisher Scientific)[38]. Raw MS data were analyzed by MaxQuant V1.6.6 software (Max Planck Institute of Biochemistry) using the Andromeda database search algorithm[39]. The spectra were searched against protein sequences downloaded from NCBI database (NC_002654.2, https://www.ncbi.nlm.nih.gov/nuccore/NC_002654.2/) for HearNPV protein identification. The iBAQ value, which represent the protein abundance normalized to the number of theoretically observable peptides, was used to estimate the absolute amounts of protein[40].

## Viral infection of *H. armigera* larvae

To collect infected hemolymph for mass spectrometry, PO activity assay, and quantitative PCR analysis, *H. armigera* larvae were infected with HearNPV by the droplet feeding method[41]. Occlusion bodies (OBs) of vHacBac-*egfp-ph*, vHacBacΔ*p26-egfp-ph*, and vHacBacRe*p26-egfp-ph* were harvested and purified from diseased larvae. Third instar *H. armigera* larvae were starved for 16 h, and then, OBs of each recombinant virus were fed to the larvae at a final concentration of $10^7$ per mL. Infected larvae were reared separately in 24-well plates and monitored daily.

The oral infectivity assay of the recombinant viruses was conducted by the droplet feeding method as described above. Larvae were fed eight concentrations ($3 \times 10^6$, $1 \times 10^6$, $3 \times 10^5$, $1 \times 10^5$, $3 \times 10^4$, $1 \times 10^4$, $3 \times 10^3$, and $1 \times 10^3$ OBs/mL) of each recombinant virus. Median lethal concentration ($LC_{50}$) values were calculated by POLO probit analysis and were compared via the lethal dose ratio method. $ST_{50}$ of the recombinant viruses was measured in third instar larvae exposed to $3 \times 10^6$ OBs/mL of each virus. Forty-eight insects were used, and the mortality was checked every 6 h. The $ST_{50}$ values were calculated using the Kaplan-Meier estimator and further compared using Log-rank test. All experiments were performed in duplicate.

## Expression of recombinant proteins in the *Drosophila* S2 cell system

For PO activity analysis, *p26/poxin* homologs including HearNPV *p26*, HearNPV *p26*^S56A, HearNPV *p26*^D58A, HearNPV *p26*^H110A, HearNPV *p26*^S56A/H110A/D58A, HearNPV *p26*^H66A, HearNPV *p26*^H66D/H67D, HearNPV *p26*^D241H/E243H, HearNPV *p26*^H66D/H67D/D241H/E243H, VACV *poxin*, AcMNPV *p26*, SfMNPV *p26*−1, SfMNPV *p26*-2, SeMNPV *p26*−1, SeMNPV *p26*-2, and *M. mediaror p26* without a signal peptide sequence were amplified from viral genomes. The PCR products were cloned into **pMT-BiP/V5-HisA** vector. The plasmids were transfected into *Drosophila* S2 cells along with **pCoHygro** hygromycin selection vector, and stable cell lines were screened according to the manufacturer's instruction of DES Inducible/ Secreted Kit (Invitrogen, K413001)[16]. The cell supernatants containing recombinant proteins were harvested and loaded onto a nickel-charged resin and eluted with imidazole. The eluted samples were concentrated for further purification using a Superdex-200

column. The purified proteins were concentrated to 10 to 20 mg/mL and stored at −80 °C before use.

## PO activity assay

PO activity of infected hemolymph was determined as previously described[15]. Briefly, 1 µL cell-free hemolymph was added to a 200-µL substrate solution (2 mM dopamine in 50 mM sodium phosphate buffer, pH 6.5) in 96-well plates. PO activity was determined by measuring the absorbance at 470 nm with a microplate reader (Synergy H1, BioTek). The values of PO activity were obtained via Gen5 software (BioTek). One unit of PO activity was defined as $\Delta A470$ of 0.001 in one minute.

To analyze the inhibitory effect of PO activation by P26/poxin homologs against naive hemolymph, cell-free hemolymph from healthy third instar *H. armigera* or *S. frugiperda* larvae was collected. Hemolymph (1 µL) was added to 9 µL of recombinant proteins (HearNPV P26 containing twin-strep tag, HearNPV P26^S56A, HearNPV P26^D58A, HearNPV P26^H110A, HearNPV P26^S56A/H110A/D58A, HearNPV P26^H66A, HearNPV P26^H66D/H67D, HearNPV P26^D241H/E243H, HearNPV P26^H66D/H67D/D241H/E243H, VACV poxin, AcMNPV P26, SfMNPV P26-1, SfMNPV P26-2, SeMNPV P26-1, SeMNPV P26-2, or *M. mediaror* P26) expressed in *Drosophila* S2 cells at final protein concentrations of 0, 1.5, 3, 4.5, and 6 mg/mL (for *H. armigera*) or 0, 3, 6, 9, and 12 mg/mL (for *S. frugiperda*). The reaction mixtures were incubated on ice for ~40 min. Then, the 10-µL reaction mixtures were subjected to the PO activity assay. BSA was used as the negative control. Each test was performed three times. Statistical analysis was performed using two-way analysis of variance (ANOVA).

## Time-course analysis of HearNPV P26

Cell-free hemolymph was collected from *H. armigera* larvae infected with vHaBac-*egfp-ph* at 0, 12, 24, 36, 48 and 72 h p.i., treated with SDS-PAGE loading buffer (50 mM Tris-HCl, 2% SDS, 0.1% bromophenol blue, 10% glycerol, 5% β-mercaptoethanol) and heated at 95 °C. Proteins were separated on a 12% SDS-PAGE and western blotting was carried out using primary polyclonal-antibodies against HearNPV P26, *H. armigera* HSP27.2[15], and HRP-conjugated secondary antibody (Affinity, S0001). All the antibody dilutions are 1:5000, and the scans were obtained using Gel Capture Microchem (DNR Bio Imaging System).

To generate antisera against HearNPV P26, the prokaryotically expressed recombinant HearNPV P26 proteins were used for rabbit immunization. The animal experiments were approved by the Institutional Animal Care and Use Committee (IACUC) of Wuhan Institute of Virology, Chinese Academy of Sciences (ethics number: WIVA01201601).

## Quantitative PCR (qPCR) analysis of genomic DNA copy numbers in hemolymph

The cell-free hemolymph from third instar larvae ($n = 5$) infected with vHacBac-*egfp-ph*, vHacBacΔ*p26-egfp-ph*, and vHacBacRe*p26-egfp-ph* were collected, and viral genomic DNA was isolated. Cell-free hemolymph (10 µL) was mixed with 90 µL $H_2O$ and incubated with 100 µL 20% PEG8000 (Sigma, V900156) in 1 M NaCl for 30 min. The virions were then spun down at $12,000 \times g$, 4 °C for 15 min and resuspended in 20 µL $H_2O$. Virions were lysed by adding 80 µL virus disruption buffer (10 mM Tris-HCl pH 7.5, 10 mM EDTA, 0.25% SDS) and 5 µL proteinase K (20 mg/mL; Sigma, P2380). Virions were lysed at 50 °C for 1 h. The viral DNA was extracted with phenol, precipitated with ethanol, and dissolved in 10 µL $H_2O$. The virus loads in hemolymph were determined via qPCR analysis of viral DNA using primers of *ha39* (*ha39*-F: 5′-GAAATGCGAATCAGACAGATTACTCG-3′ and *ha39*-R: 5′-CGCAACCTAACATTTGAGAACACAC-3′), as described previously[42]. Each test was performed twice. Data was collected using CFX manager (Bio-Rad). Results were analyzed statistically via two-tailed Student's t test.

## Pull-down assay

Purified HearNPV P26 with twin-strep tag expressed by S2 cells was incubated with Strep-Tactin XT IBA beads at 4 °C for 4 h. Then, the beads were washed five times, incubated with proSP41, procSP1, procSP6, procSPH11, or procSPH50 at 4 °C overnight. After extensive washes, the beads were resuspended in SDS-PAGE loading buffer. The samples were heated at 95 °C and then centrifuged at $12,000 \times g$ for 10 min. The supernatant was subjected to western blotting as described above, using primary antibodies against HearNPV P26, V5 tag (Abcam, ab15828) and HRP-conjugated secondary antibody. All the antibody dilutions are 1:5000. The scans were obtained using Gel Capture Microchem (DNR Bio Imaging System).

## In vitro determination of the inhibitory step in the melanization pathway

To identify the target of HearNPV P26, a biochemically reconstituted pathway was used[16]. Using procSPH11 and procSPH50 as an example, 3 μM procSPH11 and procSPH50 were mixed with 0.8 mg/mL HearNPV P26 and incubated overnight at 4 °C. Concurrently, the upstream protease procSP6 was sequentially activated by Factor Xa-proSP41xa-procSP1. Next, the activated SP6 was added to the mixture and incubated at 27 °C for 1 h. Then, 3 μM proPO was added to the mixture before determining PO activity. The rest of the groups, namely, proSP41$_{Xa}$, procSP1, and procSP6, were individually mixed with HearNPV P26 and similarly assayed for PO activity. Each test was performed three times. Results were analyzed statistically via two-tailed Student's t test.

To detect the cleavage of procSPHs, 3 μM procSPH11 or 3 μM procSPH50 was incubated with HearNPV P26, overnight at 4 °C. Then, 3 μM of activated cSP6$_{Xa}$ was added to the mixture, incubated at 27 °C for 1 h, and subjected to western blotting using primary polyclonal-antibodies against HearNPV P26, procSPH11, procSPH50 and HRP-conjugated secondary antibody. The antibody dilutions are 1:5000. The scans were obtained using Gel Capture Microchemi (DNR Bio Imaging System).

## Statistics

All of the analyses and data plotting were performed using Graph-Pad Prism v8.0.2 software. A two-tailed Student's t test or two-way ANOVA test was applied to to calculate the significance as indicated in each figure legend. The bioassay (Supplementary Table 3) was analyzed using SPSS Statistics v22 software. The LC$_{50}$ values were calculated using a POLO probit analysis and was compared via the lethal dose ratio method. The ST$_{50}$ values were calculated using a Kaplan-Meier estimator and further compared applying a Log-rank test. A $P$ value <0.05 was considered statistically significant. Error bars of all figures represent as mean ± SEM of two or three independent experiments.

## Reporting summary

Further information on research design is available in the Nature Portfolio Reporting Summary linked to this article.

## Data availability

The data generated in this study are available from the corresponding authors upon request. Atomic coordinates and structure factors for the crystal structure generated in this study have been deposited in the Protein Data bank (https://www.rcsb.org) under PDB code 7WN7. Previously published crystal structures used in this study are available from the PDB codes 6EA9, 6XB3, 6XB5 and 5GPI. The mass spectrometry proteomics data have been deposited in the ProteomeXchange Consortium with the dataset identifier PXD037784. Source data are provided with this paper.

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

## Acknowledgements

This research was supported by the grants from the National Natural Science Foundation of China (No. 32170156 to Z.H.), the Strategic Priority Research Program of the Chinese Academy of Sciences (grant XDB11030400 to Z.H. and XDPB16 to Z.Z.), Hubei Natural Science Foundation for Distinguished Young Scholar (2021CFA050 to M.W.) and the Young Top-notch Talent Cultivation Program of Hubei Province to M. W. We thank Mr. Xijia Liu for antibody preparation; Dr. Guoliang Lu and Dr. Bo Shu for their help during X-ray diffraction data collection; Dr. Guiqing Peng for kindly providing the SeMet reagent; Dr. Qiangqiang Han from SpecAlly Life Technology Co., Ltd. for his assistance with MS data analysis. We thank staff of BL17U1 and BL18U1, Shanghai Synchrotron Radiation Facility and the Center for Instrumental Analysis and Metrology of Wuhan Institute of Virology for their technical support.

## Author contributions

M.Y., M.W., Z.H., Z.Z., and P.G. designed the experiments and supervised the study; M.Y., and W.K. conducted most of experiments with the assistance from Q.W., X.W., C.Y., Z.L., H. Z., and F.D.; M.Y., W.K., and M.W. wrote the draft; Z.H., Z.Z. H.J., and P.G., revised the draft. All the authors read and approved with the final paper.

## Competing interests

The authors declare no competing interests.
