## [Peer Review File · Nature Communications]

Dual roles and evolutionary implications of P26/poxin in antagonizing intracellular cGAS-STING and extracellular melanization immunityREVIEWER COMMENTS

Reviewer #1 (Remarks to the Author):

Yin and Kuang et al. present an elegant series of biochemical, structural, and in vivo analyses that demonstrate that baculovirus poxin proteins interfere with the melanization response in lepidoptera. The experiments are thorough, well presented, and the new results are very exciting for multiple fields related to immunology and host-virus interactions. Control experiments are necessary to validate some of the findings, or alternatively the authors should correct discussion of a putative protease active site with text changes. Otherwise, I have only minor comments to help improve the manuscript for the general audience of Nature Communications.

Major Comments

1) The authors report a "putative serine protease active site" in HearNPV poxin but do not provide evidence to demonstrate protease activity. It is not clear from the current data if this is a functional protease site, and the presented data more strongly suggest that HearNPV poxin may suppress PPO activation by binding to procSPH11/procSPH50 and blocking access of upstream protease factors. The authors should consider removing references to a "putative serine protease active site" or provide substantive evidence verifying protease function. Three related points:

1A – The authors identify a putative serine protease active site S56/H110/D58, but the residues are in an atypical organization that is unlikely to support proteolysis. Trypsin and related proteases possess a triad composed of histidine, aspartate, serine in that order in the primary sequence. Notably, the residues in the HearNPV structure do not appear from the figure to be structurally compatible with the positioning necessary to form a functional trypsin-like protease active site. The authors should provide structural overlays with functional proteases, or alternatively show that these residues overlay with inactive protease active sites previously identified in lepidopteran poxins (Eaglesham et al eLife 2020 PMID 33191912).

1B – Further evidence in the manuscript suggest the HearNPV structure does not contain a functional protease active site. The authors demonstrate these putative active site residues are not evolutionary conserved within signal-peptide containing poxin proteins (Extended Data Figure 6). Additionally, mutation of the putative catalytic serine residue S56 does not impact suppression of PO activity (Figure 3e). How do the authors reconcile these findings with a putative role in protease function?

1C – Most importantly, the authors do not present any evidence of proteolytic cleavage activity by HearNPV poxin. Notably, no proteolytic activity is evident in the pull-down assays (Figure 3b). Specific experimental demonstration of proteolytic activity is necessary to verify the authors' hypothesis. Alternatively, the authors may want to consider repeating the PO activity assay in Figure 3c and use western blotting to assess if HearNPV poxin blocks proteolytic activation of procSPH11 and/or procSPH50.

Minor Comments

3) The authors do not test any lepidopteran host-encoded SP-poxins in the PO activity assay. While not required for publication it would be interesting to assess the ability of host-encoded SP poxins to inhibit PO activity and regulate melanization especially given that at least some predicted isoforms possess a signal peptide and are likely to be secreted. Figure 4c is slightly confusing as it indicates that all poxin proteins with a signal peptide are depicted as inhibiting melanization while this is not known to be true for host poxins.

4) The authors should cite and incorporate in their model figure (Figure 4) two recent papers describing cGAS-like receptors (cGLRs) as pattern recognition receptors that function upstream of STING in insects (Slavik et al Nature 2021 PMID 34261127; Holleufer et al Nature 2021 PMID 34261128). Notably, these papers demonstrate recognition of viral dsRNA but there is currently no evidence for detection of dsDNA in insects. They also report synthesis of poxin-susceptible (2'3'-cGAMP) and poxin-resistant (3'2'-cGAMP) signals.

5) Given that the SP-poxins studied here are active 2'3'-cGAMP nucleases, the authors should

discuss the known role for 2'3'-cGAMP as an important extracellular signal in mammals (Lutejin et al. Nature 2019 PMID 31511694; Ritchie et al. Molecular Cell 2019 PMID 31126740) and insects (Segrist et al Cell Reports 2021 PMID 34965418).

I hope the authors will find my comments useful, thank you for the opportunity to read this exciting manuscript.

Philip Kranzusch

Reviewer #2 (Remarks to the Author):

Summary:

The work by Yin et. al., "Dual roles and evolutionary implications of P26/poxin in innate immunity," describes the new findings that secreted viral and parasitoid P26 proteins inhibit insect melanization/phenoloxidase activity, an important part of the insect immune response. The viral P26 is secreted into the hemolymph of the infected lepidopteran insects in large amounts, which was associated with less phenoloxidase activity. They find that the inhibitory ability of the P26 derives from the protein's protease domain, and the melanin inhibition is lost following site-directed mutagenesis of the protease residues. The authors found that the P26 interacts with two components of the phenoloxidase activation pathway, serine protease homologs cSPH11 and cSPH50, and this interaction is important for inhibition to occur. The authors suggest the P26 are co-opted from the hosts' P26 genes, and thus represent a way in which the virus have harnessed the host's own genes against it. This study is fascinating, well-done, and tremendously enhances our understanding of alphabaculoviral pathogenesis in Lepidoptera. It also adds to growing and appreciation that immune melanization could have antiviral effect, which could support additional research into possible viral-melanin interactions of arboviruses and implications for human health (albeit viral-melanin interactions unrelated to P26). The authors have previously described the importance of melanization in antiviral responses in insects, and additional studies have shown that some viruses have evolved phenoloxidase-inhibiting properties and that melanization byproducts are antiviral.

The approach taken by the authors is robust, sound, and convincing in proving that viral and parasitoid wasp P26 inhibits immune melanization in Lepidoptera. The data and methodology are sound and convincing as well. I am not experienced in X-ray crystallography, so I cannot comment on those data.

Comments/Questions:

1. The experiments in this paper are well-done, the findings are extremely fascinating, and the manuscript was relatively easy and clear to read. The title could be a little more specific and mention that the P26 suppresses a key-part of the antiviral immune response.
2. I think it is typical for Nature Communications publications to have separate Introduction, Results, Discussion, and Methods sections. It would be helpful if authors separate the manuscript accordingly and include the Materials and Methods in the main manuscript text. I think separating the sections will also allow authors to include more background information on the melanization pathways and the P26/poxin family, more discussion of results and possible implications, and include more of their data (particularly like extended figure 4) in the main manuscript.
3. For the discussion, it would be helpful to include some possible hypotheses for the mechanism of inhibition.
4. Authors could describe the melanization pathways a little more, and include some description of PO/SPH Melanization Complexes (Reviewed here: https://link.springer.com/chapter/10.1007/978-3-030-41769-7_5). Is it possible that the P26 binds to SPHs and physically/sterically blocks the formation of melanization complexes?
5. The summary diagrams in Figure 3a and 4c are very helpful in helping to guide the reader in understanding the results, as is the phylogenetic tree in Figure 1b.
6. The inclusion of the red dot to label the P26 proteins that have the signal peptide was helpful for interpreting the results.

7. I think it would be helpful to include a table at the beginning of the paper listing all the viruses tested (including their abbreviation, full name, presence of a signal peptide, and the typical insect host). For example, the manuscript goes right into describing work with AcMNPV and HearNPV without defining that HearNPV is a nucleopolyhedrovirus found in *Helicoverpa armigera* and that AcMNPV means the *Autographa californica* Multiple Nucleopolyhedrovirus.
8. It would be helpful if the authors did not abbreviate signal peptide as "SP". It is not used that frequently and might cause some confusion with the serine protease and serine protease homologue proteins that are called "" and "SPH_".
9. For Extended Data Table 3, is the $\Delta p26$ and REp26 survival data switched? Based off the confidence intervals and p-values, it would seem so.
10. Nature Portfolio requires authors to show all the data points in graphs when possible prior to publishing rather than just an average with s.e.m./standard deviation/95% confidence interval. Additionally, authors should be aware of color choices for their graphs to make sure they are legible for people with colorblindness. Colorblind palette options are available on GraphPad Prism and on this website: <https://davidmathlogic.com/colorblind/#%23D81B60-%231E88E5-%23FFC107-%23004D40>
11. The Western Blots and SDS-PAGE Gels (Figure 2c, 3b, Extended Figure 2, Extended Figure 4) seem to have been either over-exposed or overprocessed to remove the background. I am sure they are not edited in a misleading way, but including the original blot with little to no processing strengthens the evidence and would be helpful to readers.

Experiment Suggestions:

12. As far as I could tell, the authors have not determined whether the lepidopteran secreted P26 gene has melanin-modulating activity. I think testing to see if the Lepidopteran P26 inhibits melanization would be helpful and demonstrate a new way in which insects can modulate their own melanization response and prevent hyper-activation.
 - o Was there a technical reason that prevented this?
 - o Is there any information of when P26 is typically expressed by the host?
 - o Does it get expressed during infection or at baseline.
 - o Would the Mass Spec differentiate between the viral and host P26?
13. From Figure 3b and Extended Figure 4, it seems as if the P26 does not cleave the cSPH11 or cSPH50, so the implication is that the mechanism of inhibition would be unrelated to direct protease activity. It also seems possible that if P26 were an active serine protease, it could result in the activation of the phenoloxidase system by cleaving the procSPs, procSPHS, and proPO directly, which could be discussed more in the paper. It would be helpful for the authors to show whether the P26 has any protease activity using a general protease activity assay, similar to what was done with the nuclease activity.

Reviewer #3 (Remarks to the Author):

In this manuscript, Yin et al. report that P26 proteins, an evolutionarily conserved set of proteins found in poxviruses, baculoviruses, cypoviruses, hymenopteran parasitic wasps, and lepidopteran insects, have two separable functions in suppressing the insect innate immune system. While an immunosuppressive function of P26 involving nuclease-mediated degradation of cellular 2,3-cGAMP had already been demonstrated, the presence of signal peptides for extracellular export in some P26 homologs led the authors to investigate a potential extracellular function of P26. The authors determined a crystal structure for a baculoviral (HearNPV) P26 protein and found putative protease and nuclease sites in distinct domains. The authors had previously reported that HearNPV inhibits the melanization immune response of *Helicoverpa armigera* (cotton bollworm) and investigated which viral proteins were responsible. The authors found that the most abundant viral protein in infected hemolymph was P26 and increasing P26 levels during infection correlated with decreased melanization activity in vivo. The authors found a dose dependent decrease in the activity of key melanization enzyme phenoloxidase (PO) with increasing HearNPV P26 levels in vitro. Infection of *H. armigera* with P26-deleted HearNPV led to increased PO activity, decreased viral titer, and increased mean time to lethality during infection compared to wild type or revertant controls. The authors found that HearNPV P26 binds to uncleaved cofactors (procSPH11 and

procSPH50) of PO activator cSP6. Pre-incubation of HearNPV P26 with procSPH11 and procSPH50 before introduction into an in vitro biochemical PO generation pathway inhibited PO activity in a dose dependent manner but did not prevent activating cleavage of these proteins by cSP6, meaning the mechanism of P26-mediated PO inhibition remains unclear. Mutations of HearNPV P26 altering the nuclease site or surface charge did not affect PO generation, while mutations affecting the serine protease site did affect PO generation without compromising nuclease activity. The authors found that all viral and hymenopteran P26 homologs tested containing a signal peptide were able to inhibit PO generation in hemolymph from two different lepidopteran insects, while P26 homologs without a signal peptide could not. HearNPV P26 most robustly inhibited *H. armigera* hemolymph PO activity while SfMNPV most robustly inhibited *S. frugiperda* hemolymph PO activity, suggesting selective evolutionary pressure. However, the serine protease motif was not conserved amongst P26 homologs, suggesting the mechanisms of PO inhibition may be diverse.

Overall, the authors demonstrate an extracellular melanization-repressive function of P26 involving inhibition of PO generation separable from nuclease activity which has in vivo consequences for the course of *H. armigera* HearNPV infection. They also demonstrate the role of the signaling peptide in mediating this mechanism across P26 proteins in diverse species.

The broader relevance of these insects and their pathogens was not very clear and could be expanded upon. Some suggestions listed below would help strengthen the paper.

- 1) The experiment in Figure 3C should be performed with pre-incubation of P26 with procSPH11 and procSPH50 separately to see how those results compare to incubation of P26 with procSPH11 and procSPH50 together. This could help illuminate if interaction with one of these cofactors contributes more to the phenotype than the interaction with the other, or if P26 must create a complex with both components simultaneously to cause PO inhibition.
- 2) Statistics are displayed on most graphs in this manuscript and should be added to Figure 2D and 3D for consistency.
- 3) The non-methods part of the manuscript should be divided into sections with Introduction, Results, and Discussion. The pathways and processes involved in melanization should be introduced earlier in the manuscript for clarity as they are the primary focus.
- 4) In line 161, the text refers to Extended Data Fig. 3a-c, but the correctly corresponding data is in Extended Data Fig. 4a-c.

REVIEWER COMMENTS

Reviewer #1 (Remarks to the Author):

Yin and Kuang et al. present an elegant series of biochemical, structural, and in vivo analyses that demonstrate that baculovirus poxin proteins interfere with the melanization response in lepidoptera. The experiments are thorough, well presented, and the new results are very exciting for multiple fields related to immunology and host-virus interactions. Control experiments are necessary to validate some of the findings, or alternatively the authors should correct discussion of a putative protease active site with text changes. Otherwise, I have only minor comments to help improve the manuscript for the general audience of Nature Communications.

Major Comments

1) The authors report a “putative serine protease active site” in HearNPV poxin but do not provide evidence to demonstrate protease activity. It is not clear from the current data if this is a functional protease site, and the presented data more strongly suggest that HearNPV poxin may suppress PPO activation by binding to procSPH11/procSPH50 and blocking access of upstream protease factors. The authors should consider removing references to a “putative serine protease active site” or provide substantive evidence verifying protease function. Three related points:

1A – The authors identify a putative serine protease active site S56/H110/D58, but the residues are in an atypical organization that is unlikely to support proteolysis. Trypsin and related proteases possess a triad composed of histidine, aspartate, serine in that order in the primary sequence. Notably, the residues in the HearNPV structure do not appear from the figure to be structurally compatible with the positioning necessary to form a functional trypsin-like protease active site. The authors should provide structural overlays with functional proteases, or alternatively show that these residues overlay with inactive protease active sites previously identified in lepidopteran poxins (Eaglesham et al eLife 2020 PMID 33191912).

1B – Further evidence in the manuscript suggest the HearNPV structure does not contain a functional protease active site. The authors demonstrate these putative active site residues are not evolutionary conserved within signal-peptide containing poxin proteins (Extended Data Figure 6). Additionally, mutation of the putative catalytic serine residue S56 does not impact suppression of PO activity (Figure 3e). How do the authors reconcile these findings with a putative role in protease function?

1C – Most importantly, the authors do not present any evidence of proteolytic cleavage activity by HearNPV poxin. Notably, no proteolytic activity is evident in the pull-down assays (Figure 3b). Specific experimental demonstration of proteolytic activity is necessary to verify the authors’ hypothesis. Alternatively, the authors may want to consider repeating the PO activity assay in Figure 3c and use western blotting to assess if HearNPV poxin blocks proteolytic activation of procSPH11 and/or procSPH50.

Minor Comments

3) The authors do not test any lepidopteran host-encoded SP-poxins in the PO activity assay. While not required for publication it would be interesting to assess the ability of host-encoded SP poxins to inhibit PO activity and regulate melanization especially given that at least some predicted isoforms possess a signal peptide and are likely to be secreted. Figure 4c is slightly confusing as it indicates that all poxin proteins with a signal peptide are depicted as inhibiting melanization while this is not known to be true for host poxins.

4) The authors should cite and incorporate in their model figure (Figure 4) two recent papers describing cGAS-like receptors (cGLRs) as pattern recognition receptors that function upstream of STING in insects (Slavik et al Nature 2021 PMID 34261127; Holleufer et al Nature 2021 PMID 34261128). Notably, these papers demonstrate recognition of viral dsRNA but there is currently no evidence for detection of dsDNA in insects. They also report synthesis of poxin-susceptible (2'3'-cGAMP) and poxin-resistant (3'2'-cGAMP) signals.

5) Given that the SP-poxins studied here are active 2'3'-cGAMP nucleases, the authors should discuss the known role for 2'3'-cGAMP as an important extracellular signal in mammals (Lutejin et al. Nature 2019 PMID 31511694; Ritchie et al. Molecular Cell 2019 PMID 31126740) and insects (Segrist et al Cell Reports 2021 PMID 34965418).

I hope the authors will find my comments useful, thank you for the opportunity to read this exciting manuscript.

Philip Kranzusch

Reviewer #2 (Remarks to the Author):

Summary:

The work by Yin et. al., “Dual roles and evolutionary implications of P26/poxin in innate immunity,” describes the new findings that secreted viral and parasitoid P26 proteins inhibit insect melanization/phenoloxidase activity, an important part of the insect immune response. The viral P26 is secreted into the hemolymph of the infected lepidopteran insects in large amounts, which was associated with less phenoloxidase activity. They find that the inhibitory ability of the P26 derives from the protein’s protease domain, and the melanin inhibition is lost following site-directed mutagenesis of the protease residues. The authors found that the P26 interacts with two components of the phenoloxidase activation pathway, serine protease homologs cSPH11 and cSPH50, and this interaction is important for inhibition to occur. The authors suggest the P26 are co-opted from the hosts’ P26 genes, and thus represent a way in which the virus have harnessed the host’s own genes against it. This study is fascinating, well-done, and tremendously enhances our understanding of alphabaculoviral pathogenesis in Lepidoptera. It also adds to growing and appreciation that immune melanization could have antiviral effect, which could support additional research into possible viral-melanin interactions of arboviruses and implications for human health (albeit

viral-melanin interactions unrelated to P26). The authors have previously described the importance of melanization in antiviral responses in insects, and additional studies have shown that some viruses have evolved phenoloxidase-inhibiting properties and that melanization byproducts are antiviral.

The approach taken by the authors is robust, sound, and convincing in proving that viral and parasitoid wasp P26 inhibits immune melanization in Lepidoptera. The data and methodology are sound and convincing as well. I am not experienced in X-ray crystallography, so I cannot comment on those data.

Comments/Questions:

1. The experiments in this paper are well-done, the findings are extremely fascinating, and the manuscript was relatively easy and clear to read. The title could be a little more specific and mention that the P26 suppresses a key-part of the antiviral immune response.
2. I think it is typical for Nature Communications publications to have separate Introduction, Results, Discussion, and Methods sections. It would be helpful if authors separate the manuscript accordingly and include the Materials and Methods in the main manuscript text. I think separating the sections will also allow authors to include more background information on the melanization pathways and the P26/poxin family, more discussion of results and possible implications, and include more of their data (particularly like extended figure 4) in the main manuscript.
3. For the discussion, it would be helpful to include some possible hypotheses for the mechanism of inhibition.
4. Authors could describe the melanization pathways a little more, and include some description of PO/SPH Melanization Complexes (Reviewed here: https://link.springer.com/chapter/10.1007/978-3-030-41769-7_5). Is it possible that the P26 binds to SPHs and physically/sterically blocks the formation of melanization complexes?
5. The summary diagrams in Figure 3a and 4c are very helpful in helping to guide the reader in understanding the results, as is the phylogenetic tree in Figure 1b.
6. The inclusion of the red dot to label the P26 proteins that have the signal peptide was helpful for interpreting the results.
7. I think it would be helpful to include a table at the beginning of the paper listing all the viruses tested (including their abbreviation, full name, presence of a signal peptide, and the typical insect host). For example, the manuscript goes right into describing work with AcMNPV and HearNPV without defining that HearNPV is a nucleopolyhedroviruses found in *Helicoverpa armigera* and that AcMNPV means the *Autographa californica* Multiple Nucleopolyhedrovirus.
8. It would be helpful if the authors did not abbreviate signal peptide as “SP”. It is not used that frequently and might cause some confusion with the serine protease and serine protease homologue proteins that are called “” and “SPH_”.
9. For Extended Data Table 3, is the $\Delta p26$ and REp26 survival data switched? Based off the confidence intervals and p-values, it would seem so.
10. Nature Portfolio requires authors to show all the data points in graphs when possible prior to publishing rather than just an average with s.e.m./standard deviation/95% confidence interval.

Additionally, authors should be aware of color choices for their graphs to make sure they are legible for people with colorblindness. Colorblind palette options are available on GraphPad Prism <https://davidmathlogic.com/colorblind/#%23D81B60-%231E88E5-%23FFC107-%23004D40>

11. The Western Blots and SDS-PAGE Gels (Figure 2c, 3b, Extended Figure 2, Extended Figure 4) seem to have been either over-exposed or overprocessed to remove the background. I am sure they are not edited in a misleading way, but including the original blot with little to no processing strengthens the evidence and would be helpful to readers.

Experiment Suggestions:

12. As far as I could tell, the authors have not determined whether the lepidopteran secreted P26 gene has melanin-modulating activity. I think testing to see if the Lepidopteran P26 inhibits melanization would be helpful and demonstrate a new way in which insects can modulate their own melanization response and prevent hyper-activation.

o Was there a technical reason that prevented this?

o Is there any information of when P26 is typically expressed by the host?

o Does it get expressed during infection or at baseline.

o Would the Mass Spec differentiate between the viral and host P26?

13. From Figure 3b and Extended Figure 4, it seems as if the P26 does not cleave the cSPH11 or cSPH50, so the implication is that the mechanism of inhibition would be unrelated to direct protease activity. It also seems possible that if P26 were an active serine protease, it could result in the activation of the phenoloxidase system by cleaving the procSPs, procSPHs, and proPO directly, which could be discussed more in the paper. It would be helpful for the authors to show whether the P26 has any protease activity using a general protease activity assay, similar to what was done with the nuclease activity.

Reviewer #3 (Remarks to the Author):

In this manuscript, Yin et al. report that P26 proteins, an evolutionarily conserved set of proteins found in poxviruses, baculoviruses, cytoviruses, hymenopteran parasitic wasps, and lepidopteran insects, have two separable functions in suppressing the insect innate immune system. While an immunosuppressive function of P26 involving nuclease-mediated degradation of cellular 2,3-cGAMP had already been demonstrated, the presence of signal peptides for extracellular export in some P26 homologs led the authors to investigate a potential extracellular function of P26. The authors determined a crystal structure for a baculoviral (HearNPV) P26 protein and found putative protease and nuclease sites in distinct domains. The authors had previously reported that HearNPV inhibits the melanization immune response of *Helicoverpa armigera* (cotton bollworm) and investigated which viral proteins were responsible. The authors found that the most abundant viral protein in infected hemolymph was P26 and increasing P26 levels during infection correlated with decreased melanization activity in vivo. The authors found a dose dependent decrease in the activity of key melanization enzyme phenoloxidase (PO) with increasing HearNPV P26 levels in vitro. Infection of *H. armigera* with

P26-deleted HearNPV led to increased PO activity, decreased viral titer, and increased mean time to lethality during infection compared to wild type or revertant controls. The authors found that HearNPV P26 binds to uncleaved cofactors (procSPH11 and procSPH50) of PO activator cSP6. Pre-incubation of HearNPV P26 with procSPH11 and procSPH50 before introduction into an in vitro biochemical PO generation pathway inhibited PO activity in a dose dependent manner but did not prevent activating cleavage of these proteins by cSP6, meaning the mechanism of P26-mediated PO inhibition remains unclear. Mutations of HearNPV P26 altering the nuclease site or surface charge did not affect PO generation, while mutations affecting the serine protease site did affect PO generation without compromising nuclease activity. The authors found that all viral and hymenopteran P26 homologs tested containing a signal peptide were able to inhibit PO generation in hemolymph from two different lepidopteran insects, while P26 homologs without a signal peptide could not. HearNPV P26 most robustly inhibited *H. armigera* hemolymph PO activity while SfMNPV most robustly inhibited *S. frugiperda* hemolymph PO activity, suggesting selective evolutionary pressure. However, the serine protease motif was not conserved amongst P26 homologs, suggesting the mechanisms of PO inhibition may be diverse.

Overall, the authors demonstrate an extracellular melanization-repressive function of P26 involving inhibition of PO generation separable from nuclease activity which has in vivo consequences for the course of *H. armigera* HearNPV infection. They also demonstrate the role of the signaling peptide in mediating this mechanism across P26 proteins in diverse species. The broader relevance of these insects and their pathogens was not very clear and could be expanded upon. Some suggestions listed below would help strengthen the paper.

- 1) The experiment in Figure 3C should be performed with pre-incubation of P26 with procSPH11 and procSPH50 separately to see how those results compare to incubation of P26 with procSPH11 and procSPH50 together. This could help illuminate if interaction with one of these cofactors contributes more to the phenotype than the interaction with the other, or if P26 must create a complex with both components simultaneously to cause PO inhibition.
- 2) Statistics are displayed on most graphs in this manuscript and should be added to Figure 2D and 3D for consistency.
- 3) The non-methods part of the manuscript should be divided into sections with Introduction, Results, and Discussion. The pathways and processes involved in melanization should be introduced earlier in the manuscript for clarity as they are the primary focus.
- 4) In line 161, the text refers to Extended Data Fig. 3a-c, but the correctly corresponding data is in Extended Data Fig. 4a-c.

REVIEWER COMMENTS

Reviewer #1 (Remarks to the Author):

Yin and Kuang et al. present an elegant series of biochemical, structural, and in vivo analyses that demonstrate that baculovirus poxin proteins interfere with the melanization response in lepidoptera. The experiments are thorough, well presented, and the new results are very exciting for multiple fields related to immunology and host-virus interactions. Control experiments are necessary to validate some of the findings, or alternatively the authors should correct discussion of a putative protease active site with text changes. Otherwise, I have only minor comments to help improve the manuscript for the general audience of Nature Communications.

Response: We appreciate the reviewer very much for the positive comments and valuable suggestions.

Major Comments

1) The authors report a “putative serine protease active site” in HearNPV poxin but do not provide evidence to demonstrate protease activity. It is not clear from the current data if this is a functional protease site, and the presented data more strongly suggest that HearNPV poxin may suppress PPO activation by binding to procSPH11/procSPH50 and blocking access of upstream protease factors. The authors should consider removing references to a “putative serine protease active site” or provide substantive evidence verifying protease function. Three related points:

1A – The authors identify a putative serine protease active site S56/H110/D58, but the residues are in an atypical organization that is unlikely to support proteolysis. Trypsin and related proteases possess a triad composed of histidine, aspartate, serine in that order in the primary sequence. Notably, the residues in the HearNPV structure do not appear from the figure to be structurally compatible with the positioning necessary to form a functional trypsin-like protease active site. The authors should provide structural overlays with functional proteases, or alternatively show that these residues overlay with inactive protease active sites previously identified in lepidopteran poxins (Eaglesham et al eLife 2020 PMID 33191912).

1B – Further evidence in the manuscript suggest the HearNPV structure does not contain a functional protease active site. The authors demonstrate these putative active site residues are not evolutionary conserved within signal-peptide containing poxin proteins (Extended Data Figure 6). Additionally, mutation of the putative catalytic serine residue S56 does not impact suppression of PO activity (Figure 3e). How do the authors reconcile these findings with a putative role in protease function?

1C – Most importantly, the authors do not present any evidence of proteolytic cleavage activity by HearNPV poxin. Notably, no proteolytic activity is evident in the pull-down assays (Figure 3b). Specific experimental demonstration of proteolytic activity is necessary to verify the authors’ hypothesis. Alternatively, the authors may want to consider repeating the PO activity assay in Figure 3c and use western blotting to assess if HearNPV poxin blocks proteolytic

activation of procSPH11 and/or procSPH50.

Response: We thank the reviewer very much for the critical comments and we agree with the reviewer's points. When the crystal structure of HearNPV P26 was solved, we performed homologous structure search and found that some hits to viral proteases (please see the Table below), and identified a putative serine protease-like triad in HearNPV P26. These hints led to the surmise that HearNPV P26 might contain a protease domain. But as the reviewer commented, the residues don't occur in a typical histidine, aspartate, and serine primary sequence order. And the serine protease-like triad is present in only sporadic P26 homologs (**Extended Data Fig. 4**).

Table | Homologous structure search of HearNPV P26

No	PDB ID	Z-score	rmsd (Å)	Sequence identity (%)	Homologs
1	6XB3	31.3	1.7	36	Autographa californica nucleopolyhedrovirus poxin
2	6XB5	18.0	2.6	15	Trichoplusia ni poxin
3	6XB6	17.8	2.8	13	Danaus plexippus poxin
4	6EA8	17.4	2.9	26	Vaccinia virus WR poxin
5	6XB4	16.6	3.2	13	Pieris rapae granulovirus poxin
6	5YVW	9.6	6.2	11	Dengue virus 4 NS3 protease
8	5GPI	9.1	3.2	9	Zika virus NS3 protease
9	2FP7	9.0	3.3	6	West Nile virus NS3 protease
10	6B6I	8.7	3.5	9	human Norovirus GI.4 3C-like protease
11	4X2V	8.4	3.5	10	Murine norovirus 1 NS6 Protease

Following the reviewer's suggestion, we conducted structural overlays of HearNPV P26 vs Zika virus NS3 protease (with functional protease active site-residues of H51, D75, and S135) and HearNPV P26 vs lepidopteran *T. ni* poxin (with mutated active site-residues of H41, D60, and A120). The results showed that the protease triad of HearNPV P26 doesn't overlap with that of Zika virus NS3 protease or *T. ni* poxin (see **Figure** below, which is also included as **Extended Data Fig. 5**).

Figure: Structure overlays of HearNPV P26 versus *T. ni* Poxin (left) and HearNPV P26 versus Zika virus NS3 protease (right).

In addition, as pointed by the reviewer, the mutation of the putative catalytic serine residue S56 showed only minor impact on suppression of PO activity (**Fig. 5c**). And most importantly, we failed to show proteolytic cleavage activity (**Fig. 4**). Taken together, we think it is unlikely that HearNPV P26 acts through a functional protease domain. We have revised the relevant descriptions and discussions (**Abstract, line 44; Results, lines 120 and 234; Discussion, lines 306–328**).

Minor Comments

3) The authors do not test any lepidopteran host-encoded SP-poxins in the PO activity assay. While not required for publication it would be interesting to assess the ability of host-encoded SP poxins to inhibit PO activity and regulate melanization especially given that at least some predicted isoforms possess a signal peptide and are likely to be secreted. Figure 4c is slightly confusing as it indicates that all poxin proteins with a signal peptide are depicted as inhibiting melanization while this is not known to be true for host poxins.

Response: We thank the reviewer for the suggestion. This is certainly an interesting question and we are planning to investigate the function of the lepidopteran poxins in PPO activation. We have checked previous proteomic and transcriptomic data and found some interesting preliminary information (please see Response to Reviewer 2's comment #12). We have revised the model figure (**the current Fig. 7**) by adding a **gray background** and a **question mark** to the lepidopteran poxins, and explained this in the figure legend. We have also added some discussion in the main text (**lines 273–283**).

4) The authors should cite and incorporate in their model figure (Figure 4) two recent papers describing cGAS-like receptors (cGLRs) as pattern recognition receptors that function upstream of STING in insects (Slavik et al Nature 2021 PMID 34261127; Holleufer et al Nature 2021 PMID 34261128). Notably, these papers demonstrate recognition of viral dsRNA but there is currently no evidence for detection of dsDNA in insects. They also report synthesis of poxin-susceptible (2'3'-cGAMP) and poxin-resistant (3'2'-cGAMP) signals.

Response: We thank the reviewer for the valuable suggestion. The two important papers have now been cited in the text (**lines 285–298**). In addition, we have revised the model figure (**the current Fig. 7**) by incorporating cGLRs and poxin-resistant (3'2'-cGAMP) signals. We believe the new model more comprehensively represents the current knowledge on the interplay between pathogens and host cGAS/STING immunity.

5) Given that the SP-poxins studied here are active 2'3'-cGAMP nucleases, the authors should discuss the known role for 2'3'-cGAMP as an important extracellular signal in mammals (Lutejin et al. Nature 2019 PMID 31511694; Ritchie et al. Molecular Cell 2019 PMID 31126740) and insects (Segrist et al Cell Reports 2021 PMID 34965418).

Response: Indeed, the extracellular 2'3'-cGAMP functions as an important immunotransmitter in mammals and insects. We have added a paragraph to discuss this possibility in the Discussion (**lines 300–304**) and have revised the model figure (**Fig. 7**) accordingly.

I hope the authors will find my comments useful, thank you for the opportunity to read this exciting manuscript.

Response: We are very grateful to the reviewer for all the insightful suggestions, and we think the modifications have significantly improved the quality of the manuscript.

Philip Kranzusch

Reviewer #2 (Remarks to the Author):

Summary:

The work by Yin et. al., “Dual roles and evolutionary implications of P26/poxin in innate immunity,” describes the new findings that secreted viral and parasitoid P26 proteins inhibit insect melanization/phenoloxidase activity, an important part of the insect immune response. The viral P26 is secreted into the hemolymph of the infected lepidopteran insects in large amounts, which was associated with less phenoloxidase activity. They find that the inhibitory ability of the P26 derives from the protein’s protease domain, and the melanin inhibition is lost following site-directed mutagenesis of the protease residues. The authors found that the P26 interacts with two components of the phenoloxidase activation pathway, serine protease homologs cSPH11 and cSPH50, and this interaction is important for inhibition to occur. The authors suggest the P26 are co-opted from the hosts’ P26 genes, and thus represent a way in which the virus have harnessed the host’s own genes against it.

This study is fascinating, well-done, and tremendously enhances our understanding of alphabaculoviral pathogenesis in Lepidoptera. It also adds to growing and appreciation that immune melanization could have antiviral effect, which could support additional research into possible viral-melanin interactions of arboviruses and implications for human health (albeit viral-melanin interactions unrelated to P26). The authors have previously described the importance of melanization in antiviral responses in insects, and additional studies have shown that some viruses have evolved phenoloxidase-inhibiting properties and that melanization byproducts are antiviral.

The approach taken by the authors is robust, sound, and convincing in proving that viral and parasitoid wasp P26 inhibits immune melanization in Lepidoptera. The data and methodology are sound and convincing as well. I am not experienced in X-ray crystallography, so I cannot comment on those data.

Response: We appreciate the reviewer very much for the positive comments and useful suggestions.

Comments/Questions:

1. The experiments in this paper are well-done, the findings are extremely fascinating, and the

manuscript was relatively easy and clear to read. The title could be a little more specific and mention that the P26 suppresses a key-part of the antiviral immune response.

Response: Thanks. The title has been revised as “*Dual roles and evolutionary implications of P26/poxin in antagonizing intracellular cGAS-STING and extracellular melanization immunity.*”

2. I think it is typical for Nature Communications publications to have separate Introduction, Results, Discussion, and Methods sections. It would be helpful if authors separate the manuscript accordingly and include the Materials and Methods in the main manuscript text. I think separating the sections will also allow authors to include more background information on the melanization pathways and the P26/poxin family, more discussion of results and possible implications, and include more of their data (particularly like extended figure 4) in the main manuscript.

Response: We thank the reviewer for the suggestion. The manuscript was previously transferred from *Nature* via the transfer system directly, and now we have modified the manuscript format to meet with the requirements of *Nature Communications* in the revision. Following reviewers’ suggestions, more introduction and discussion on melanization and cGAS-STING pathway, are provided in the revision (**Introduction, lines 77–86; Discussion, lines 285–304 and 306–328**). In addition, we have incorporated extended Figure 4 (**the new Fig. 4**) and extended Figure 5 (grouped with the previous Fig. 3e and f as **the new Fig. 5**) into the main manuscript. Moreover, the Fig. 4 of the previous version has been divided into **Fig. 6** and **Fig. 7** to make it easier to follow.

3. For the discussion, it would be helpful to include some possible hypotheses for the mechanism of inhibition.

Response: As suggested by the same reviewer in the next question, we agree that P26 binds to SPHs and physically/sterically blocks the formation of melanization complexes, which needs further investigations. We have added this possibility in the Discussion (**lines 316–328**).

4. Authors could describe the melanization pathways a little more, and include some description of PO/SPH Melanization Complexes (Reviewed here: https://link.springer.com/chapter/10.1007/978-3-030-41769-7_5). Is it possible that the P26 binds to SPHs and physically/sterically blocks the formation of melanization complexes?

Response: We thank the reviewer for the suggestion, and relevant description of PO/SPH melanization complexes has been added (**lines 316–323**). As mentioned above, we also agree with the reviewer that binding of P26 to SPHs may cause sterically hindrance, and this possibility is discussed in the revision (**lines 323–328**).

5. The summary diagrams in Figure 3a and 4c are very helpful in helping to guide the reader in understanding the results, as is the phylogenetic tree in Figure 1b.

Response: Thanks for the positive comments.

6. The inclusion of the red dot to label the P26 proteins that have the signal peptide was helpful for interpreting the results.

Response: Thanks.

7. I think it would be helpful to include a table at the beginning of the paper listing all the viruses tested (including their abbreviation, full name, presence of a signal peptide, and the typical insect host). For example, the manuscript goes right into describing work with AcMNPV and HearNPV without defining that HearNPV is a nucleopolyhedrovirus found in *Helicoverpa armigera* and that AcMNPV means the *Autographa californica* Multiple Nucleopolyhedrovirus.

Response: We thank the reviewer for the suggestion. A summary table (**Table 1**) with related information regarding all the viruses tested has been added at the beginning of the paper (**Results, line 103**).

8. It would be helpful if the authors did not abbreviate signal peptide as “SP”. It is not used that frequently and might cause some confusion with the serine protease and serine protease homologue proteins that are called “” and “SPH_”.

Response: We thank the reviewer for the suggestion. We have carefully checked the text and figures, and replaced all the abbreviations “SP” by the full name “signal peptide”.

9. For Extended Data Table 3, is the $\Delta p26$ and REp26 survival data switched? Based off the confidence intervals and p-values, it would seem so.

Response: The data were not switched. The LC_{50} of the $\Delta p26$ and REp26 were not statistically different, but the LT_{50} of $\Delta p26$ were significantly longer than that of the Rep26 and WT. The P values were based on the log-rank test, and the conclusion was in accordance with the judgments by the overlap of confidence intervals.

10. Nature Portfolio requires authors to show all the data points in graphs when possible prior to publishing rather than just an average with s.e.m./standard deviation/95% confidence interval. Additionally, authors should be aware of color choices for their graphs to make sure they are legible for people with colorblindness. Colorblind palette options are available on GraphPad Prism and on this website:

<https://davidmathlogic.com/colorblind/#%23D81B60-%231E88E5-%23FFC107-%23004D40>

Response: We thank the reviewer for the suggestion. We have shown all the data points in the graphs and modified the color scheme according to the Journal’s requirements.

11. The Western Blots and SDS-PAGE Gels (Figure 2c, 3b, Extended Figure 2, Extended Figure 4) seem to have been either over-exposed or overprocessed to remove the background. I am sure they are not edited in a misleading way, but including the original blot with little to no processing strengthens the evidence and would be helpful to readers.

Response: We understand the concern of the reviewer. The original blots and gels with minimal processing have been provided in the revised **Figure 2c, 3b, the new Figure 4, and Extended Figure 2.**

Experiment Suggestions:

12. As far as I could tell, the authors have not determined whether the lepidopteran secreted P26 gene has melanin-modulating activity. I think testing to see if the Lepidopteran P26 inhibits melanization would be helpful and demonstrate a new way in which insects can modulate their own melanization response and prevent hyper-activation.

o Was there a technical reason that prevented this?

o Is there any information of when P26 is typically expressed by the host?

o Does it get expressed during infection or at baseline.

o Would the Mass Spec differentiate between the viral and host P26?

Response: We agree with the reviewer that the function of Lepidopteran P26 is a very interesting topic and in fact it deserves a deep investigation. We have checked our previous proteomic and transcriptomic data, and found some interesting preliminary information:

1) *H. armigera* P26 (HaP26) was induced in midgut but not in other tissues by baculovirus HearNPV infection (Yuan et al., *Insect Science*, 2021, 28:1766-1779; NCBI SRA: PRJNA386740; <https://www.ncbi.nlm.nih.gov/sra/?term=PRJNA386740>).

2) HaP26 was significantly up-regulated in fat bodies after fungi (*Beauveria bassiana*) challenge (Table S4 in Xiong et al., *BMC Genomics*, 2015,16:321).

3) HaP26 was also induced in whole body sample after larvae were parasitized by wasp *Microplitis mediator* (NCBI SRA: PRJNA349107; <https://www.ncbi.nlm.nih.gov/sra/?term=PRJNA349107>).

These data suggested that HaP26 (signal peptide-containing) is an immune responsive gene induced by a wide variety of pathogens. Therefore, whether the secreted lepidopteran poxins play a role in regulating melanization pathway deserves further investigation, and such studies would also shine lights onto how P26/poxin proteins are evolved during long time adaption between insect host and pathogens. The relevant discussions have been added in the revision (273–283). So far, we have not tested whether HaP26 inhibits melanization, as we have not yet developed a S2 cell line stably expressing HaP26.

13. From Figure 3b and Extended Figure 4, it seems as if the P26 does not cleave the cSPH11 or cSPH50, so the implication is that the mechanism of inhibition would be unrelated to direct protease activity. It also seems possible that if P26 were an active serine protease, it could result in the activation of the phenoloxidase system by cleaving the procSPs, procSPHs, and proPO

directly, which could be discussed more in the paper. It would be helpful for the authors to show whether the P26 has any protease activity using a general protease activity assay, similar to what was done with the nuclease activity.

Response: We see the reviewer's point. Please also see our response to **Reviewer 1 (the Major Comments #1)**. Although HearNPV P26 contains a serine protease-like triad, the residues don't occur in a typical organization (in the order of histidine, aspartate, serine in the primary sequence). And the serine protease-like triad are present in only sporadic P26 homologs (**Extended Data Fig. 4**). The protease-like triad of HearNPV P26 doesn't overlap with that of a functional protease Zika virus NS3 protease or a degenerated *T. ni* poxin (**Extended Data Fig. 5**). In addition, the putative catalytic serine residue S56 showed only minor impact on suppression of PO activity (the new **Fig. 5c**). Most importantly, we failed to show proteolytic cleavage activity (**the new Fig. 4**). Thus, it is unlikely that P26 functions through a protease activity. We assume that the mechanism could be that P26 binds to SPHs and physically blocks the formation of melanization complexes.

We have revised relevant descriptions and discussions in the revision (**Abstract, line 44; Results, lines 120 and 234; Discussion, lines 306–328**).

Reviewer #3 (Remarks to the Author):

In this manuscript, Yin et al. report that P26 proteins, an evolutionarily conserved set of proteins found in poxviruses, baculoviruses, cytopoviruses, hymenopteran parasitic wasps, and lepidopteran insects, have two separable functions in suppressing the insect innate immune system. While an immunosuppressive function of P26 involving nuclease-mediated degradation of cellular 2,3-cGAMP had already been demonstrated, the presence of signal peptides for extracellular export in some P26 homologs led the authors to investigate a potential extracellular function of P26. The authors determined a crystal structure for a baculoviral (HearNPV) P26 protein and found putative protease and nuclease sites in distinct domains. The authors had previously reported that HearNPV inhibits the melanization immune response of *Helicoverpa armigera* (cotton bollworm) and investigated which viral proteins were responsible. The authors found that the most abundant viral protein in infected hemolymph was P26 and increasing P26 levels during infection correlated with decreased melanization activity in vivo. The authors found a dose dependent decrease in the activity of key melanization enzyme phenoloxidase (PO) with increasing HearNPV P26 levels in vitro. Infection of *H. armigera* with P26-deleted HearNPV led to increased PO activity, decreased viral titer, and increased mean time to lethality during infection compared to wild type or revertant controls. The authors found that HearNPV P26 binds to uncleaved cofactors (procSPH11 and procSPH50) of PO activator cSP6. Pre-incubation of HearNPV P26 with procSPH11 and procSPH50 before introduction into an in vitro biochemical PO generation pathway inhibited PO activity in a dose dependent manner but did not prevent activating cleavage of these proteins by cSP6, meaning the mechanism of P26-mediated PO inhibition remains unclear. Mutations of HearNPV P26 altering the nuclease site or surface charge did not affect PO generation, while mutations

affecting the serine protease site did affect PO generation without compromising nuclease activity. The authors found that all viral and hymenopteran P26 homologs tested containing a signal peptide were able to inhibit PO generation in hemolymph from two different lepidopteran insects, while P26 homologs without a signal peptide could not. HearNPV P26 most robustly inhibited *H. armigera* hemolymph PO activity while SfMNPV most robustly inhibited *S. frugiperda* hemolymph PO activity, suggesting selective evolutionary pressure. However, the serine protease motif was not conserved amongst P26 homologs, suggesting the mechanisms of PO inhibition may be diverse.

Overall, the authors demonstrate an extracellular melanization-repressive function of P26 involving inhibition of PO generation separable from nuclease activity which has in vivo consequences for the course of *H. armigera* HearNPV infection. They also demonstrate the role of the signaling peptide in mediating this mechanism across P26 proteins in diverse species. The broader relevance of these insects and their pathogens was not very clear and could be expanded upon. Some suggestions listed below would help strengthen the paper.

Response: We thank the reviewer for the careful review and very helpful comments.

1) The experiment in Figure 3C should be performed with pre-incubation of P26 with procSPH11 and procSPH50 separately to see how those results compare to incubation of P26 with procSPH11 and procSPH50 together. This could help illuminate if interaction with one of these cofactors contributes more to the phenotype than the interaction with the other, or if P26 must create a complex with both components simultaneously to cause PO inhibition.

Response: We see the point of the reviewer. As demonstrated in our previous study, procSPH11 and procSPH50 of *H. armigera* need to function synergistically, and each individual lacks obvious PO-enhancing activity (Wang *et al.*, 2020, *Front Immunol*; PMID 32431706). Similar situation was found in other insect system (Wang *et al.*, 2014, *Insect Biochem Mol Biol*; PMID 24768974). That is why SPH combinations are called as “a cofactor” instead of “cofactors”. Based on this background information, we did not perform the experiments of pre-incubation of P26 with procSPH11 and procSPH50 separately, which is not expected to set up an effective PPO activation system for identifying the effect of P26. We have added some background information about SPHs as a cofactor in the text of revision (**lines 181–183; 316–324**).

2) Statistics are displayed on most graphs in this manuscript and should be added to Figure 2D and 3D for consistency.

Response: Thanks, and we have added statistics to **Figure 2d** and **3d**.

3) The non-methods part of the manuscript should be divided into sections with Introduction, Results, and Discussion. The pathways and processes involved in melanization should be introduced earlier in the manuscript for clarity as they are the primary focus.

Response: We thank the reviewer for the suggestion. As described in the response to **Reviewer 2**, the manuscript was previously transferred from *Nature*, and we have now modified the format to meet with the requirements of *Nature Communications*. In addition, more background introduction and discussion have been added to improve the clarity of the manuscript.

4) In line 161, the text refers to Extended Data Fig. 3a-c, but the correctly corresponding data is in Extended Data Fig. 4a-c.

Response: We thank the reviewer for the kind reminder. The mistake has been corrected.

References:

1. Eaglesham, J.B. et al. Structures of diverse poxin cGAMP nucleases reveal a widespread role for cGAS-STING evasion in host–pathogen conflict. *Elife* **9**, e59753 (2020).
2. Holleufer, A., Winther, K. G., and Gad, H.H. et al. Two cGAS-like receptors induce antiviral immunity in *Drosophila*. *Nature*. **597**, 114-118 (2021).
3. Lutejin, R.D. and Zaver, S.A. et al. SLC19A1 transports immunoreactive cyclic dinucleotides. *Nature*. **573**, 434-438 (2019).
4. Ritchie, C. et al. SLC19A1 is an importer of the immunotransmitter cGAMP. *Mol. Cell*. **75**, 372-381 (2019).
5. Segrist, E. et al. Orally acquired cyclic dinucleotides drive dSTING-dependent antiviral immunity in enterocytes. *Cell Rep*. **37**, 110150 (2021).
6. Slavik, K.M. et al. cGAS-like receptors sense RNA and control 3'2'-cGAMP signalling in *Drosophila*. *Nature*. **597**, 109-113 (2021).
7. Wang, Q. et al. Identification of a conserved prophenoloxidase activation pathway in cotton bollworm *Helicoverpa armigera*. *Front. Immunol*. **11**, 785-796 (2020).
8. Wang, Y. et al. *Manduca sexta* prophenoloxidase activating proteinase-3 (PAP3) stimulates melanization by activating proPAP3, proSPHs, and proPOs. *Insect Biochem Mol Biol* **50**, 82-91 (2014).
9. Xiong, G., Xing, L. et al. High throughput profiling of the cotton bollworm *Helicoverpa armigera* immunotranscriptome during the fungal and bacterial infections. *BMC Genomics* **16**, 321 (2015).
10. Yuan, C. et al. Microbiota modulates gut immunity and promotes baculovirus infection in *Helicoverpa armigera*. *Insect Sci* **28**, 1766-1779 (2021).

REVIEWERS' COMMENTS

Reviewer #1 (Remarks to the Author):

The authors' revised manuscript is significantly improved. All of my reviewer points have been addressed, and I congratulate the authors on a very exciting story.

Reviewer #2 (Remarks to the Author):

The authors have sufficiently addressed all of the suggested edits, and I have no further comments or suggestions.

Reviewer #3 (Remarks to the Author):

The authors have addressed my prior concerns.

REVIEWERS' COMMENTS

Reviewer #1 (Remarks to the Author):

The author' revised manuscript is significantly improved. All of my reviewer points have been addressed, and I congratulate the authors on a very exciting story.

Reviewer #2 (Remarks to the Author):

The authors have sufficiently addressed all of the suggested edits, and I have no further comments or suggestions.

Reviewer #3 (Remarks to the Author):

The authors have addressed my prior concerns.

REVIEWERS' COMMENTS

Point-to-point reply: We sincerely thank the three reviewers for the careful review and valuable comments.

Reviewer #1 (Remarks to the Author):

The author's revised manuscript is significantly improved. All of my reviewer points have been addressed, and I congratulate the authors on a very exciting story.

Reviewer #2 (Remarks to the Author):

The authors have sufficiently addressed all of the suggested edits, and I have no further comments or suggestions.

Reviewer #3 (Remarks to the Author):

The authors have addressed my prior concerns.